# Private Non-smooth ERM and SCO in Subquadratic Steps

Janardhan Kulkarni[*]      Yin Tat Lee [†]      Daogao Liu [‡]

## Abstract

We study the differentially private Empirical Risk Minimization (ERM) and Stochastic Convex Optimization (SCO) problems for non-smooth convex functions. We get a (nearly) optimal bound on the excess empirical risk for ERM with $O(\frac{N^{3/2}}{d^{1/8}} + \frac{N^2}{d})$ gradient queries, which is achieved with the help of subsampling and smoothing the function via convolution. Combining this result with the iterative localization technique of Feldman et al. [FKT20], we achieve the optimal excess population loss for the SCO problem with $O(\min\{N^{5/4}d^{1/8}, \frac{N^{3/2}}{d^{1/8}}\})$ gradient queries. Our work makes progress towards resolving a question raised by Bassily et al. [BFGT20], giving first algorithms for private SCO with subquadratic steps. In a concurrent work, Asi et al. [AFKT21] gave other algorithms for private ERM and SCO with subquadratic steps.

## 1 Introduction

Privacy has become an important consideration for learning algorithms dealing with sensitive data. Over the past decade, differential privacy, introduced in the seminal work of [DMNS06], has established itself as the defacto notion of privacy for machine learning problems. In this paper, we revisit Empirical Risk Minimization (ERM) and Stochastic Convex Optimization (SCO) problems, two fundamental problems in statistics and machine learning, in differential privacy setting. In the ERM problem, we are given a family of convex functions $\{f(\cdot, x)\}_{x \in \Xi}$ over a bounded closed convex set $\mathcal{K} \subset \mathbb{R}^d$ of diameter $D$, a data set $S = \{x_1, \cdots, x_N\}$ drawn from some unknown distribution $\mathcal{P}$ over the universe $\Xi$, and the objective is to

$$\text{minimize} \quad \widehat{F}(\omega) := \frac{1}{N} \sum_{x_i \in S} f(\omega, x_i) \quad \text{over} \quad \omega \in \mathcal{K},$$

The excess empirical loss with respect to a solution $\omega$ is defined by $\widehat{F}(\omega) - \widehat{F}^*$, where $\widehat{F}^* = \min_{\omega \in \mathcal{K}} \widehat{F}(\omega)$. A closely related but more general problem is SCO, where the objective is to

$$\text{minimize} \quad F(\omega) := \mathop{\mathbb{E}}_{x \sim \mathcal{P}} f(\omega, x) \quad \text{over} \quad \omega \in \mathcal{K},$$

We refer $F(\omega) - F^*$ as the *excess population loss*, where $F(\omega) = \mathbb{E}_{x \sim \mathcal{P}} f(\omega, x)$ and $F^* = \min_{\omega \in \mathcal{K}} F(\omega)$.

Differentially private convex optimization has been studied extensively for over a decade now [CM08, RBHT09, CMS11, KST12, JT14, TTZ14, BST14, TTZ15, KJ16, WLK⁺17, FTS17,

---

[*]Algorithms Group, Microsoft Research (MSR) Redmond. `jakul@microsoft.com`

[†]University of Washington and Microsoft Research. Supported by NSF awards CCF-1749609, DMS-1839116, DMS-2023166, Microsoft Research Faculty Fellowship, Sloan Research Fellowship, Packard Fellowships.`yintat@uw.edu`

[‡]University of Washington. Part of the work was done while visiting Shanghai Qi Zhi Institute.`dgliu@uw.edu`.

35th Conference on Neural Information Processing Systems (NeurIPS 2021).

ZZMW17, WYX17, INS$^+$19]. Both DP-ERM and DP-SCO for smooth convex functions are well understood in the sense that we know (near) linear time algorithms that achieve optimal loss; we refer the readers to [WYX17, FKT20] for more details. However, for the more general non-smooth convex loss functions our understanding is not yet complete, which is the focus of this paper.

Our algorithms for DP-SCO build upon our improvements for the DP-ERM problem, and hence we begin with DP-ERM problem. A summary of the state-of-the-art results of DP-ERM and our contributions for the non-smooth convex loss functions is given in Table 1. We will discuss the concurrent work [AFKT21] separately at the end of the introduction, and the following discussion is only limited to the previous work.

| | General Convex | Strongly Convex | Gradient Complexity |
|---|---|---|---|
| [KST12] | $\frac{GD\sqrt{d}\log(1/\delta)}{\sqrt{N}\varepsilon}$ | $\frac{G^2 d\log(1/\delta)}{\mu N^{3/2}\varepsilon^2}$ | N/A |
| [BST14] | $\frac{GD\log^{\frac{3}{2}}(N/\delta)\sqrt{d\log(1/\delta)}}{N\varepsilon}$ | $\frac{G^2\log^2(N/\delta)d\log(1/\delta)}{\mu N^2\varepsilon^2}$ | $N^2$ |
| [BFTT19] | $\frac{GD\sqrt{d\log(1/\delta)}}{N\varepsilon}$ | $\frac{G^2 d\log(1/\delta)}{\mu N^2\varepsilon^2}$ | $N^{4.5}$ |
| [BFGT20] | $\frac{GD\sqrt{d\log(1/\delta)}}{N\varepsilon}$ | $\frac{G^2 d\log(1/\delta)}{\mu N^2\varepsilon^2}$ | $N^2$ |
| [AFKT21] | $\frac{GD\sqrt{d\log(1/\delta)}}{N\varepsilon}$ | $\frac{G^2 d\log(1/\delta)}{\mu N^2\varepsilon^2}$ | $N^2/\sqrt{d}$ |
| Ours | $\frac{GD\sqrt{d\log(1/\delta)}}{N\varepsilon}$ | $\frac{G^2 d\log(1/\delta)}{\mu N^2\varepsilon^2}$ | $\frac{N^{3/2}}{d^{1/8}} + \frac{N^2}{d}$ |

Table 1: DP-ERM. Comparisons with previous $(\varepsilon, \delta)$-differential private algorithms when objective function is $G$-Lipschitz and convex (or $\mu$-strongly convex) over a convex set $\mathcal{K} \subset \mathbb{R}^d$ of diameter $D$. The results are stated asymptotically and the big $O$ notation is hidden for simplicity. The lower bound of General Convex function is $\Omega(\min\{GD, \frac{GD\sqrt{d}}{N\varepsilon}\})$ and of Strongly Convex function is $\Omega(\min\{\frac{G^2}{\mu}, \frac{G^2 d}{\mu N^2\varepsilon^2}\})$ [BST14].

[KST12] used objective perturbation method to design a DP-algorithm with $O(\frac{GD\sqrt{d}\log(1/\delta)}{\sqrt{N}\varepsilon})$ excess empirical risk. This result was improved significantly by [BST14], who first showed a lower bound of $\Omega(\min\{GD, \frac{GD\sqrt{d}}{N\varepsilon}\})$ on the excess empirical risk for DP-ERM. Further, they gave an algorithm with excess empirical risk $O(\frac{GD\log^{\frac{3}{2}}(N/\delta)\sqrt{d\log(1/\delta)}}{N\varepsilon})$, which is sub-optimal by a factor of $\log^{\frac{3}{2}}(N/\delta)$. Their algorithm is based on a modification of SGD by adding Gaussian noise to the gradients to make it differentially private. The privacy analysis proceeds via amplification by sampling and the strong composition theorem. Roughly speaking, the logarithmic blowup in the excess empirical risk is due to two reasons: 1) The strong composition theorem requires that at each step one needs to add Gaussian noise with a larger variance; 2) They used sub-optimal convergence rate $O(\log T/\sqrt{T})$ for $T$-step SGD.

However, getting the optimal bounds with small gradient complexity for non-smooth case turns out to be a more difficult problem. This was noted by [WYX17], who raised it as an important open problem. This question was answered in [BFTT19], who gave an algorithm with almost optimal excess empirical risk. To achieve this, [BFTT19] first consider the smooth case, and give an improved privacy analysis via the Moments Accountant technique proposed by [ACG$^+$16]. They extend their result to non-smooth case by applying Moreau-Yosida envelope technique (a.k.a. Moreau envelope smoothing) [Nes05] to make the function smooth. However, this technique is computationally inefficient and leads to $O(N^{4.5})$-gradient computations for the whole algorithm. This limitation was overcome in a recent work of [BFGT20] who gave the optimal excess empirical risk guarantee with $O(N^2)$-gradient computations. The privacy analysis of this result also used Moments Accountant method, and they used the standard online-to-batch conversion technique [CBCG04] to prove the high-probability and expectation bound on the excess empirical error of SGD.

As we can see from Table 1, all the previously known results (except the concurrent work [AFKT21]) achieving near optimal excess empirical risk bounds require at least $O(N^2)$-gradient computations. As Table 2 shows, a similar situation arises in Stochastic Convex Optimization (SCO), which is a closely related problem compared to ERM. Many results for SCO [BST14, BFTT19, BFGT20] are directly based on ERM; that is, solving the ERM and analyzing the generalization error. The first non-trivial result for general convex loss functions achieving excess population loss of $O\left(GD(\frac{d^{1/4}}{\sqrt{N}} + \frac{\sqrt{d}}{N\varepsilon})\right)$ was given by [BST14], who showed the result by first solving the ERM prob-

lem and bounding the generalization error. They used the result on universal convergence directly, namely, bounding $\sup_{\omega \in \mathcal{K}} \mathbb{E}[F(\omega) - \widehat{F}(\omega)]$. But this method has its limitations; For example, [Fel16] showed that lower bound of universal convergence is $\Omega(\sqrt{d/N})$ for some (not necessarily convex) loss functions. Later, [BFTT19], [FKT20] and [BFGT20] obtained near optimal excess population loss with significantly better running times (gradient complexity). The privacy analysis in these papers relied on recent advances in the privacy techniques such as the Moments Accountant method [ACG⁺16], Rényi differential privacy (RDP) [Mir17] and the Privacy Amplification by Iteration [FMTT18] and other fast stochastic convex optimization algorithms such as [JNN19]. The excess population loss bound in most of these works followed by solving a (phased) convex (regularized) ERM problem and then appealing to the uniform stability property [HRS16] or the iterative localization approach [FKT20] to do the generalization error analysis.

|  | General Convex | Strongly Convex | Gradient Complexity |
|---|---|---|---|
| [BST14] | $GD(\frac{d^{1/4}\log(n/\delta)}{\sqrt{N}} + \frac{d^{1/2}\log^2(n/\delta)}{N\varepsilon})$ | N/A | $N^2$ |
| [BFTT19] | $GD(\frac{1}{\sqrt{N}} + \frac{\sqrt{d\log(1/\delta)}}{N\varepsilon})$ | $\frac{G^2}{\mu}(\frac{1}{N} + \frac{d\log(1/\delta)}{N^2\varepsilon^2})$ | $N^{4.5}$ |
| [FKT20] | $GD(\frac{1}{\sqrt{N}} + \frac{\sqrt{d\log(1/\delta)}}{N\varepsilon})$ | $\frac{G^2}{\mu}(\frac{1}{N} + \frac{d\log(1/\delta)}{N^2\varepsilon^2})$ | $N^2\log(1/\delta)$ |
| [BFGT20] | $GD(\frac{1}{\sqrt{N}} + \frac{\sqrt{d\log(1/\delta)}}{N\varepsilon})$ | $\frac{G^2}{\mu}(\frac{1}{N} + \frac{d\log(1/\delta)}{N^2\varepsilon^2})$ | $N^2$ |
| [AFKT21] | $GD(\frac{1}{\sqrt{N}} + \frac{\sqrt{d\log(1/\delta)}}{N\varepsilon})$ | $\frac{G^2}{\mu}(\frac{1}{N} + \frac{d\log(1/\delta)}{N^2\varepsilon^2})$ | $\min\{N^{3/2}, N^2/\sqrt{d}\}$ |
| Ours | $GD(\frac{1}{\sqrt{N}} + \frac{\sqrt{d\log(1/\delta)}}{N\varepsilon})$ | $\frac{G^2}{\mu}(\frac{1}{N} + \frac{d\log(1/\delta)}{N^2\varepsilon^2})$ | $\min\{N^{5/4}d^{1/8}, \frac{N^{3/2}}{d^{1/8}}\}$ |

Table 2: DP-SCO. Comparisons with previous $(\varepsilon, \delta)$-differential private algorithms when objective function is $G$-Lipschitz and convex (or $\mu$-strongly convex) over a convex set $\mathcal{K} \subset \mathbb{R}^d$ of diameter $D$. The results are stated asymptotically and the big $O$ notation is hidden for simplicity. The lower bound of General Convex function is $\Omega(GD(\frac{1}{\sqrt{N}} + \frac{\sqrt{d}}{N\varepsilon}))$ and of Strongly Convex function is $\Omega(\frac{G^2}{\mu}(\frac{1}{N} + \frac{d\log(1/\delta)}{N^2\varepsilon^2}))$ [BST14].

Despite these impressive improvements, as the Table 2 suggests, the previous algorithms that achieve the optimal excess population loss still require $O(N^2)$-gradient computations. Indeed, [BFGT20] write that " Proving that quadratic running time is necessary for general non-smooth DP-SCO is a very interesting open problem...". Understanding if the lower bound is the right answer to the above questions or one can design algorithms with subquadratic gradient complexity is the main motivation that spurred our work.

## 1.1 Our Contributions

The first contribution of this paper is to show that we can obtain subquadratic gradient complexity bound for ERM when the dimension is super constant. In particular, for the important regime of over-parameterization ($d \geq N$), we achieve a bound of $O(N^{1+3/8})$. Let $\mathcal{K}_r = \{y \mid y = \omega + z, \omega \in \mathcal{K}, z \in \mathbb{R}^d, \|z\|_r \leq r\}$. We now state the result formally.

**Theorem 1.1** (DP-ERM). *Suppose $\mathcal{K} \subset \mathbb{R}^d$ is a closed convex set and $\{f(\cdot, x)\}_{x \in \Xi}$ is a family of $G$-Lipschitz and convex functions over $\mathcal{K}_r$, where $r = \frac{D\sqrt{d\log(1/\delta)}}{\varepsilon N}$[4]. For $\varepsilon, \delta \leq 1/2$, given any sample set $S$ consists of $N$ samples from $\Xi$ and arbitrary initial point $\omega_0 \in \mathcal{K}$, we have a $(\varepsilon, \delta)$-differentially private algorithm $\mathcal{A}$ which takes*

$$O\left(\frac{\varepsilon N^{\frac{3}{2}}}{d^{1/8}\log^{1/4}(1/\delta)} + \frac{\varepsilon^2 N^2}{d\log(1/\delta)}\right)$$

*gradient queries and outputs $\omega_T$ such that $\mathbb{E}[\widehat{F}(\omega_T) - \widehat{F}^*] = O\left(\frac{GD\sqrt{d\log(1/\delta)}}{\varepsilon N}\right)$, where $D = \|\omega^* - \omega_0\|_2, \widehat{F}(\omega) = \frac{1}{N}\sum_{x_i \in S} f(\omega, x_i), \widehat{F}^* = \min_{\omega \in \mathcal{K}} \widehat{F}(\omega)$, and the expectation is taken over the randomness of the algorithm.*

---

[4]We only need consider the non-trivial case when $\frac{\sqrt{d\log(1/\delta)}}{\varepsilon N} \leq 1$, or any feasible solution is good enough. This means that $r = O(D)$, which is a mild assumption.

*Moreover, if $\{f(\cdot, x)\}_{x \in \Xi}$ is also $\mu$-strongly convex over $\mathcal{K}_r$, we can meet the same gradient query complexity and get a solution $\omega_T$ such that $\mathbb{E}[\widehat{F}(\omega_T) - \widehat{F}^*] = O\left(\frac{G^2 d \log(1/\delta)}{\mu \varepsilon^2 N^2}\right)$.*

The main contribution of the paper is to use the above result to obtain a better gradient complexity for the SCO problem, answering the open problem in [BFGT20]. Combining our private ERM algorithm with the iterative localization technique [FKT20], we give the first algorithm achieving optimal error with strictly sub-quadratic steps for all dimensions.

**Theorem 1.2** (DP-SCO). *Suppose $\varepsilon, \delta \leq \frac{1}{2}$ and sample set $S$ consists of $N$ samples drawn i.i.d from a distribution $\mathcal{P}$ over $\Xi$. Suppose $\{f(\cdot, x)\}_{x \in \Xi}$ is convex and $G$-Lipschitz with respect to $\ell_2$ norm and convex over $\mathcal{K}_r$, where $r = \frac{D\sqrt{d \log(1/\delta)}}{\varepsilon N}$, there is an $(\varepsilon, \delta)$-differentially private algorithm taking*

$$O(N + \min\{\sqrt{\varepsilon}N^{5/4}d^{1/8}, \frac{\varepsilon N^{3/2}}{d^{1/8}\log^{1/4}(1/\delta)}\})$$

*gradient queries to get a solution $\omega_T$ such that $\mathbb{E}[F(\omega_T) - F(\omega^*)] = O(GD(\frac{1}{\sqrt{N}} + \frac{\sqrt{d \log(1/\delta)}}{N\varepsilon}))$. Moreover, if $\{f(\cdot, x)\}_{x \in \Xi}$ is also $\mu$-strongly convex over $\mathcal{K}_r$, we can meet the same gradient query complexity and get a solution $\omega_T$ such that $\mathbb{E}[F(\omega_T) - F(\omega^*)] = O\left(\frac{G^2}{\mu}(\frac{d \log(1/\delta)}{\varepsilon^2 N^2} + \frac{1}{N})\right)$.*

Finally, we note that with straightforward modifications, our results can also capture the regularized ERM and SCO, where there is one more simple (and convex) function $h(\omega)$ added to the objective function and the objective function takes the form $\frac{1}{N}\sum_{x_i \in S} f(\omega, x_i) + h(\omega)$, which show up often in the previous work such as [RBHT09, KST12, WYX17, INS+19].

## 1.2 Our Techniques

Most of the previous works [BST14, BFTT19, BFGT20] that achieve near optimal bounds for ERM and SCO are based on adaptations of SGD to make it differentially private. The information theoretic lower bound of $\Omega(1/\sqrt{T})$ for $T$-step SGD may be one of the important reasons why we can not get subquadratic gradient complexity for non-smooth convex ERM easily. Consider the algorithm in [BFGT20] as an example. It needs to add Gaussian noise $v \sim \mathcal{N}(0, \sigma^2 I_{d \times d})$ with $\sigma^2 = \frac{G^2 \log(1/\delta)}{\varepsilon^2}$ to each gradient. By a standard analysis of SGD, we can only show an excess empirical risk of $\Theta(\frac{D\sqrt{d\sigma^2}}{\sqrt{T}})$, which requires us to set $T = \Omega(N^2)$ to get ideal bound, thus hitting the quadratic barrier.

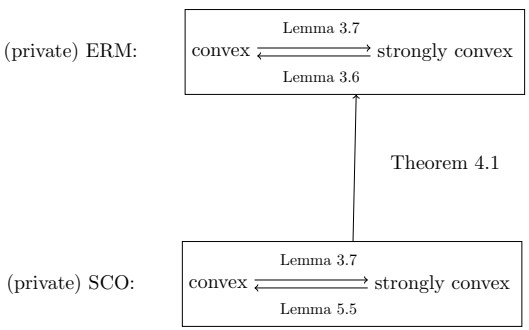

Figure 1: Reductions between ERM and SCO for general convex and strongly convex cases. Lemma 5.5 is in the full version in the supplementary material

We deviate from the above approaches for designing private algorithms for non-smooth functions. First notice that the gradient complexity $O(\frac{\varepsilon N^{\frac{3}{2}}}{d^{1/8}\log^{1/4}(1/\delta)} + \frac{\varepsilon^2 N^2}{d \log(1/\delta)})$ in Theorem 1.1 is the same for both strongly convex and general non-smooth functions; same holds for DP-SCO. This is not a coincidence; We prove that if we can achieve optimal empirical risk (population loss) for one case, then we can achieve optimal empirical risk (population loss) for another with the same privacy guarantee and gradient complexity. The Figure 1 illustrates the relationship among these different problems.

Our result for the general convex non-smooth case is obtained by providing a reduction to the strongly convex non-smooth case. Thus, our task becomes designing better algorithms for the strongly convex non-smooth functions. Rather than using SGD, we let the objective function take convolution with a sphere kernel to make it smooth. We then use the accelerated stochastic approximation algorithm in [GL12] for solving strongly convex stochastic optimization problems. However, this is not enough, as the required noise that needs to be added to the gradients to make the algorithm private is too large

to get subquadratic gradient complexity, even if we use the tighter Moments Accountant technique [ACG+16]. We overcome this by increasing the batch size to an appropriate value. Combining these ideas together, we show that the amount of noise we add can be reduced to achieve the optimal excess empirical loss, and we get the gradient complexity of $O(\max\{N^{3/2}/d^{1/8}, N^2/d\})$.

For SCO, we get the gradient complexity of $O(\min\{N^{5/4}d^{1/8}, N^{3/2}/d^{1/8}\})$ via application of the iterative localization approach of Feldman et al [FKT20]. The intuition behind iterative localization is using private ERM to solve regularized objective functions which have low sensitivity, iteration by iteration. Each iteration reduces the distance to an approximate minimizer by a multiplicative factor, so after logarithmic number of phases we are done.

### 1.3 Concurrent and Independent Work

In an independent and concurrent work, [AFKT21] give a new analysis of private regularized mirror descent to do the private ERM. Then they combine the iterative localization approach to achieve the optimal excess population loss for SCO. Their result also achieves subquaratic gradient complexity.

More formally, they get $O(\log N \cdot \min(N^{3/2}\sqrt{\log d}, N^2/\sqrt{d}))$ for SCO in query complexity. We compare their gradient complexity with ours in the figure to right. As we see, our result is better in the important regime $d \leq N^{1+1/3}$. Same holds true for ERM. Finally, we remark that the main motivation of [AFKT21] was to study SCO problem in more general $\ell_p$ norms as much of the literature has focused on the $\ell_2$-norm. They also give new results in $\ell_p$-bounded domain together with another concurrent work [BGN21].

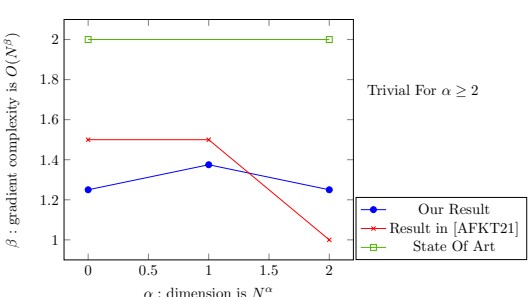

## 2 Preliminaries

We recall some basic definitions in convex optimization and DP.

**Definition 2.1** (*L*-Lipschitz Continuity). A function $f : \mathcal{K} \to \mathbb{R}$ is *L*-Lipschitz continuous over the domain $\mathcal{K} \subset \mathbb{R}^d$ if the following holds for all $\omega, \omega' \in \mathcal{K} : |f(\omega) - f(\omega')| \leq L\|\omega - \omega'\|_2$.

**Definition 2.2** ($\beta$-Smoothness). A function $f : \mathcal{K} \to \mathbb{R}$ is $\beta$-smooth over the domain $\mathcal{K} \subset \mathbb{R}^d$ if for all $\omega, \omega' \in \mathcal{K}, \|\nabla f(\omega) - \nabla f(\omega')\|_2 \leq \beta\|\omega - \omega'\|_2$.

**Definition 2.3** ($\mu$-Strongly convex). A differentiable function $f : \mathcal{K} \to \mathbb{R}$ is called strongly convex with parameter $\mu > 0$ if the following inequality holds for all points $\omega, \omega' \in \mathcal{K}, \langle \nabla f(\omega) - \nabla f(\omega'), \omega - \omega' \rangle \geq \mu\|\omega - \omega'\|_2^2$. Equivalently, $f(\omega') \geq f(\omega) + \nabla f(\omega)^\top(\omega' - \omega) + \frac{\mu}{2}\|\omega' - \omega\|_2^2$.

**Definition 2.4** (Differential privacy). A randomized mechanism $\mathcal{M}$ is $(\varepsilon, \delta)$-differentially private if for any event $\mathcal{O} \in \mathrm{Range}(\mathcal{M})$ and for any neighboring databases $S, S'$ that differ in a single data element, one has

$$\Pr[\mathcal{M}(S) \in \mathcal{O}] \leq \exp(\varepsilon)\Pr[\mathcal{M}(S') \in \mathcal{O}] + \delta.$$

### 2.1 A Meta Algorithm for DP Convex Optimization

Many DP convex optimization algorithms with noisy first-order information have the following simple format.

---

**Algorithm 1:** Private Meta Algorithm META$_{\mathsf{DP}}$

---

1  **Input:** Sample set $S = \{x_1, \cdots, x_N\}$, the objective convex function $F(\omega)$ we want to minimize, the initial point $\omega_0$, and privacy parameters $\varepsilon, \delta$;

2  **Process: for** *phases* $t = 1, \cdots, T$ **do**

3  $\quad$ Select a random sample set $S_t$ from the uniform distribution over all subsets of $S$ of size $B$;

4  $\quad$ Let $G_t = (\sum_{x_i \in S_t} \nabla f(\omega_{t-1}, x_i) + v)/B$, where $v \sim \mathcal{N}(0, \sigma^2 I_{d \times d})$;

5  $\quad$ Update the result by some sub-procedure $\omega_t \leftarrow$ Sub-procedure$(\omega_{t-1}, G_t)$;

6  **end**

7  **Output:** Some function of $\{\omega_i\}_{i \geq 1}$.

---

Compared to non-private algorithms, DP algorithms make two simple modifications to make it private. First, we compute gradients over a uniform sample of some size $B$. Next, we add a carefully calibrated Gaussian noise to these gradients and take average, before updating our results. The DP analysis then follows from a careful accounting of the privacy budget lost in each iteration, and the bound on excess empirical risk comes from the property of the optimization algorithm. The privacy analysis of the above algorithm can be done via moments account [ACG$^+$16] or using the framework of sub-sampled Gaussian mechanism, for which we can use tCDP proposed in [BDRS18]. As this is a direct application of the main result in [BDRS18], we leave the proof of the following theorem in the full version attached as supplementary material.

**Theorem 2.5.** *Suppose $\{f(\cdot, x)\}_{x \in \Xi}$ is a family of G-Lipschitz and convex functions over $\mathcal{K}$, for $\varepsilon < c_1 B^2 T / N^2$, $B \leq N/10$ and $1/2 \geq \delta > 0$, by setting $\sigma = \frac{c_2 G B \sqrt{T \log(1/\delta)}}{\varepsilon N}$ for some constant $c_1$ and $c_2$, META$_{\mathsf{DP}}$ is $(\varepsilon, \delta)$-differential private.*

# 3  Differentially Private ERM

In this section, we present private algorithms achieving the optimal excess empirical loss with subquadratic gradient complexity when the dimension is super constant. We consider non-smooth strongly-convex functions first, and then show how to reduce the general non-smooth case to the strongly-convex case in the last subsection. Due to space constraints, we won't be able to provide all the proofs in the main body; the supplementary material contains the full version of the paper with all the proofs.

## 3.1  Non-smooth Strongly-convex Functions

We use the framework introduced in Section 2.1 and give a faster private algorithm. Specifically, we modify a stochastic convex optimization algorithm (AC-SA) in [GL12] to fit into our framework.

---

**Algorithm 2:** Accelerated stochastic approximation (AC-SA) algorithm

---

1  **Input:** Initial point $\omega_0 \in \mathcal{K}$.

2  **Initialization:** Set the initial point $\omega_0^{ag} = \omega_0$;

3  Set the step-size parameters $\alpha_t = \frac{2}{t+2}$ and $\gamma_t = \frac{4L}{t(t+1)}$;

4  **Process:**

5  **for** $t = 1, \cdots, T$ **do**

6  $\quad$ Let $\omega_t^{md} = \frac{(1-\alpha_t)(\mu + \gamma_t)}{\gamma_t + (1-\alpha_t^2)\mu} \omega_{t-1}^{ag} + \frac{\alpha_t[(1-\alpha_t)\mu + \gamma_t]}{\gamma_t + (1-\alpha_t^2)\mu} \omega_{t-1}$;

7  $\quad$ Query Oracle $\mathcal{G}_t \equiv \mathcal{G}(\omega_t^{md})$;

8  $\quad$ $\omega_t = \arg\min_{\omega \in \mathcal{K}} \{\alpha_t[\langle \mathcal{G}_t, \omega \rangle + h(\omega) + \mu \|\omega_t^{md} - \omega\|_2^2] + [(1-\alpha_t)\mu + \gamma_t]\|\omega_{t-1} - \omega\|_2^2\}$;

9  $\quad$ $\omega_t^{ag} = \alpha_t \omega_t + (1-\alpha_t)\omega_{t-1}^{ag}$;

10  **end**

11  **Return:** $\omega_T^{ag}$.

---

First we recall some properties of the algorithm AC-SA. Suppose $f : \mathcal{K} \to \mathbb{R}$ is a convex function, and the objective is to get $\Psi^* := \min_{\omega \in \mathcal{K}}\{\Psi(\omega) = f(\omega) + h(\omega)\}$, where $\mathcal{K}$ is a closed convex set and $h(\omega)$ is a simple convex function with known structure.

**Theorem 3.1** (Proposition 9 in [GL12])**.** *If the following conditions are met:*

- *For some $L \geq 0, M \geq 0$ and $\mu > 0$, $\frac{\mu}{2}\|y - \omega\|_2^2 \leq f(y) - f(\omega) - \langle g(\omega), y - \omega \rangle \leq \frac{L}{2}\|y - \omega\|_2^2 + M\|y - \omega\|_2$, $\quad \forall \omega, y \in \mathcal{K}$, where $g(\omega) \in \partial f(\omega)$ and $\partial f(\omega)$ denotes the sub-differential of $f$ at $\omega$.*
- *For each call of the stochastic oracle $\mathcal{G}$ with the input $\omega_t \in \mathcal{K}$, the stochastic oracle $\mathcal{G}$ can output an independent vector $\mathcal{G}(\omega_t)$ such that $\mathbb{E}[\mathcal{G}(\omega_t)] \in \partial f(\omega_t)$.*
- *For any $t \geq 1$ and $\omega_t \in \mathcal{K}$, $\mathbb{E}[\|\mathcal{G}(\omega_t) - g(\omega_t)\|_2^2] \leq V$.*

*Then, AC-SA (algorithm 2) outputs $\omega_T$ after $T$ iterations such that $\mathbb{E}[\Psi(\omega_T) - \Psi^*] \leq O\left(\frac{L\|\omega_0 - \omega^*\|_2^2}{T^2} + \frac{M^2 + V}{\mu T}\right)$, where $\omega^* = \arg\min_{\omega \in \mathcal{K}} \Psi(\omega)$ and $\Psi^* = \Psi(\omega^*)$.*

### 3.1.1 Smoothing Function

From the statement of Theorem 3.1, it is clear that the algorithm in [GL12] gives much better convergence rates for smooth functions. As we are considering non-smooth functions, we need an efficient way to smooth the objective function without introducing too much error. In the next few paragraphs, we show how to achieve that. Recall that $D$ denotes the diameter of the closed convex set $\mathcal{K} \subset \mathbb{R}^d$. Suppose $\{f(\cdot, x)\}_{x \in \Xi}$ is a family of $G$-Lipschitz and $\mu$-strongly convex functions over $\mathcal{K}$. This implies that for any sample set $S$, the empirical loss function $\widehat{F}(\omega)$ we consider is $G$-Lipschitz and $\mu$-strongly convex over the domain $\mathcal{K}$. We do a convolution on $f(\cdot, x)$, which is denoted by $f(\cdot, x) * n_r$. The objective function after the convolution step becomes $\widehat{F}_{n_r}(\omega) = \frac{1}{N}\sum_{x_i \in S} \mathbb{E}_{y \sim n_r} f(\omega + y, x_i)$, where $n_r$ is the uniform density on the $\ell_2$ ball of radius $r$. We need the following claim:

**Claim 3.2.** *Suppose $\{f(\cdot, x)\}_{x \in \Xi}$ is convex and $G$-Lipschitz over $\mathcal{K} + B_2(0, r)$. For $\omega \in \mathcal{K}$, $\widehat{F}_{n_r}(\omega)$ has following properties:*
*1) $\widehat{F}(\omega) \leq \widehat{F}_{n_r}(\omega) \leq \widehat{F}(\omega) + Gr$;*
*2) $\widehat{F}_{n_r}(\omega)$ is $G$-Lipschitz;*
*3) $\widehat{F}_{n_r}(\omega)$ is $\frac{G\sqrt{d}}{r}$-Smooth;*
*4) For random variables $y \sim n_r$ and $x$ uniformly from $S$, one has $\mathbb{E}[\nabla f(\omega + y, x)] = \nabla \widehat{F}_{n_r}(\omega)$ and $\mathbb{E}[\|\nabla \widehat{F}_{n_r}(\omega) - \nabla f(\omega + y, x)\|_2^2] \leq G^2$.*

The properties 1)-3) of this claim come from Lemma 7 and Lemma 8 in [YNS12] while the forth follows from Lemma E.2 in [DBW12]. Furthermore, the convolution operation preserves strong convexity, which implies the fact below.

**Fact 3.3.** *Let $n_r$ be the uniform density on the $\ell_2$ ball of radius $r$, and $f : \mathcal{K}_r \to \mathbb{R}$ be a $\mu$-strongly convex function over $\mathcal{K}_r$. Then $\mathbb{E}_{y \sim n_r} f(y + \cdot)$ is $\mu$-strongly convex over $\mathcal{K}$.*

### 3.1.2 Algorithm

Recall that $y \sim n_r$ is a $d$-dimension vector drawn from the uniform density on the $\ell_2$ ball of radius $r$. Our algorithm is described in Algorithm 3 below, which is a modification of Algorithm 2.

---
**Algorithm 3:** Private $\mathsf{AC-SA}$

---
1 **Input:** A convex set $\mathcal{K}$ with diameter $D$, a family $\{f(\cdot, x_i)\}_{i \in [N]}$ of $G$-Lipschitz and $\mu$-strongly convex functions over $\mathcal{K}$, an initial point $\omega_0 \in \mathcal{K}$, privacy parameters $\varepsilon, \delta$, the batch size $B$, and the number of steps $T$.

2 Set $r \leftarrow \frac{D}{Td^{1/4}}$ and $\sigma \leftarrow \Theta(\frac{GB\sqrt{T\log(1/\delta)}}{\varepsilon N})$;
3 Run the $\mathsf{AC-SA}$ with the Oracle $\mathcal{G}$ defined below;
4 **Return:** The output of $\mathsf{AC-SA}$

5 **Oracle $\mathcal{G}(\omega)$:**
6 Select a random sample set $S_t$ from the uniform distribution over all subsets of $S$ of size $B$.
7 **Return:** $\left(\sum_{x_i \in S_t} \partial f(\omega + y_i, x_i) + v\right)/B$, where $y_i \sim n_r$ for each $i \in [B]$ and $v \sim \mathcal{N}(0, \sigma^2 \mathbf{I}_{d \times d})$.

---

### 3.1.3 Utility and Privacy

It is not hard to show that Private $\mathsf{AC-SA}$ (Algorithm 3) is an instance of $\mathsf{META_{DP}}$ (see Section 2.1), so we have the following guarantee directly by Theorem 2.5.

**Lemma 3.4.** *For $\varepsilon \leq c_1 B^2 T/N^2, \delta \leq 1/2, B \leq N/10$ and $\sigma = \frac{c_2 GB\sqrt{T\log(1/\delta)}}{\varepsilon N}$ where $c_1 \leq 1, c_2 \geq 1$ are constants, Private $\mathsf{AC-SA}$ is $(\varepsilon, \delta)$-DP.*

Now we consider the accuracy of Private $\mathsf{AC-SA}$, which is proved in the full version.

**Lemma 3.5.** *Under the assumptions defined in Algorithm Private $\mathsf{AC-SA}$, after $T$ iterations, it outputs $\omega_T$ such that*

$$\mathbb{E}[\widehat{F}(\omega_T) - \widehat{F}^*] = O\left(\frac{G^2/B + \sigma^2 d/B^2}{\mu T} + \frac{GDd^{1/4}}{T}\right),$$

*where $\omega^* = \arg\min_{\omega \in \mathcal{K}} \widehat{F}(\omega)$, and $\widehat{F}^* = \min_\omega \widehat{F}(\omega)$.*

Before stating the main result of this section, we need the following lemma that removes the dependence on the diameter term. Recall that the lower bound of strongly convex case is $\Omega(\min\{\frac{G^2}{\mu}, \frac{G^2 d\log(1/\delta)}{\mu\varepsilon^2 N^2}\})$ while for the general case is $\Omega(\min\{GD, \frac{GD\sqrt{d\log(1/\delta)}}{\varepsilon N}\})$. Therefore, we only need to think about the case when $\frac{d\log(1/\delta)}{\varepsilon^2 N^2} \leq 1$, or the bound will be trivial. The following lemma says if we can achieve sum of these two lower bounds for strongly-convex case, then we can achieve the optimal bound for the strongly-convex case, which implies we can reduce the Strongly-Convex Case to General Convex Case. The following lemma follows from the reduction in Section 5.1 in [FKT20], and we try to give a more formal statement for convenience in the future.

**Lemma 3.6** (Reduction to General Convex Case). *Given $\widehat{F}$ is $G$-Lipschitz and $\mu$-strongly convex. Suppose for any $\varepsilon, \delta < 1/2$, we have an $(\varepsilon, \delta)$-differentially private algorithm $\mathcal{A}$ which takes $\omega_0$ as the initial start point and outputs a solution $\omega_T$ such that $\mathbb{E}[\widehat{F}(\omega_T) - \widehat{F}^*] = O\left(\frac{G^2 d\log(1/\delta)}{\mu\varepsilon^2 N^2} + \frac{GD\sqrt{d\log(1/\delta)}}{\varepsilon N}\right)$, where $\omega^* = \arg\min_{\omega \in \mathcal{K}} \widehat{F}(\omega)$ and $D = \|\omega_0 - \omega^*\|_2$. Then by taking $\mathcal{A}$ as sub-procedure with some modifications on parameters, we can get an $(\varepsilon, \delta)$-differentially private solution with excess empirical loss at most $\mathbb{E}[\widehat{F}(\omega_T) - \widehat{F}^*] = O\left(\frac{G^2 d\log(1/\delta)}{\mu\varepsilon^2 N^2}\right)$. Furthermore, if $\mathcal{A}$ uses $g(N, \varepsilon, \delta)$ many gradients, the new algorithm uses $\sum_{i\geq 1} g(N, \varepsilon/2^i, \delta/2^i)$ many gradients.*

We give the proof in the full version. Now we are ready to prove the bounds for strongly convex case of Theorem 1.1.

*Proof of Theorem 1.1.* By Lemma 3.5, the output $\omega$ of Private $\mathsf{AC-SA}$ satisfies $\mathbb{E}[\widehat{F}(\omega) - \widehat{F}^*] = O\left(\frac{\frac{G^2}{B} + \frac{\sigma^2 d}{B^2}}{\mu T} + \frac{GDd^{1/4}}{T}\right)$. Setting $\sigma = \frac{c_2 GB\sqrt{T\log(1/\delta)}}{\varepsilon N}$ and $T = \lceil \frac{100\varepsilon N}{c_1 d^{1/4}\sqrt{\log(1/\delta)}} \rceil$ ($c_1, c_2$ are defined in Lemma 3.4), one has

$$\mathbb{E}[\widehat{F}(\omega) - \widehat{F}^*] = O\left(\frac{G^2}{\mu BT} + \frac{G^2 d\log(1/\delta)}{\mu\varepsilon^2 N^2} + \frac{GDd^{1/4}}{T}\right) = O\left(\frac{G^2}{\mu BT} + \frac{G^2 d\log(1/\delta)}{\mu\varepsilon^2 N^2} + \frac{GD\sqrt{d\log(1/\delta)}}{\varepsilon N}\right)$$

To ensure that Private $\mathsf{AC-SA}$ is $(\varepsilon, \delta)$-DP, we set $B = \lceil \sqrt{\frac{\varepsilon N^2}{c_1 T}} + \frac{\varepsilon^2 N^2}{d\log(1/\delta)T} \rceil$. By our choice of $T$, we have $B \leq N/10$ and $\varepsilon \leq c_1 B^2 T/N^2$. Hence, we can apply Lemma 3.4 to conclude the guarantee of $(\varepsilon, \delta)$ differential privacy. Furthermore, we get a solution $\omega$ such that

$$\mathbb{E}[\widehat{F}(\omega) - \widehat{F}^*] = O\left(\frac{G^2 d\log(1/\delta)}{\mu\varepsilon^2 N^2} + \frac{GD\sqrt{d\log(1/\delta)}}{\varepsilon N}\right).$$

As for the total gradient complexity of our algorithm, we are under the assumption that $\frac{d\log(1/\delta)}{\varepsilon^2 N^2} \leq 1$, which means that $\frac{\varepsilon N}{d^{1/4}\sqrt{\log(1/\delta)}} \geq d^{1/4}$, and $T = \lceil \frac{100\varepsilon N}{c_1 d^{1/4}\sqrt{\log(1/\delta)}} \rceil = \Theta(\frac{\varepsilon N}{d^{1/4}\sqrt{\log(1/\delta)}})$. As for

the batch size, we know $\sqrt{\frac{\varepsilon N^2}{T}} + \frac{\varepsilon^2 N^2}{d \log(1/\delta) T} = \omega(1)$ and thus $B = \lceil \sqrt{\frac{\varepsilon N^2}{T}} + \frac{\varepsilon^2 N^2}{d \log(1/\delta) T} \rceil = \Theta(\sqrt{\frac{\varepsilon N^2}{T}} + \frac{\varepsilon^2 N^2}{d \log(1/\delta) T})$, from which we get the gradient complexity is

$$BT = \Theta \left( \frac{\varepsilon N^{\frac{3}{2}}}{d^{1/8} \log^{1/4}(1/\delta)} + \frac{\varepsilon^2 N^2}{d \log(1/\delta)} \right).$$

By Lemma 3.6 we can adjust Private AC−SA and get a final solution $\omega_T$ such that

$$\mathbb{E}[\widehat{F}(\omega_T) - \widehat{F}^*] = O \left( \frac{G^2 d \log(1/\delta)}{\mu \varepsilon^2 N^2} \right),$$

with gradient complexity $\Theta(\sum_{i=1}^{\log \log N^3} \frac{(\varepsilon/2^i) N^{3/2}}{d^{1/8} \log^{1/4}(2^i/\delta)} + \frac{(\varepsilon/2^i)^2 N^2}{d \log(2^i/\delta)}) = \Theta(\frac{\varepsilon N^{\frac{3}{2}}}{d^{1/8} \log^{1/4}(1/\delta)} + \frac{\varepsilon^2 N^2}{d \log(1/\delta)})$, which completes the proof. □

### 3.2 General Non-smooth Convex Functions

In the general non-smooth case, we only assume that the family of functions $\{f(\cdot, x)\}_{x \in \Xi}$ is $G$-Lipschitz and convex over $\mathcal{K}$. We now give a reduction from this case to the strongly-convex case, which completes our main result for ERM.

**Lemma 3.7.** *Suppose $\mathcal{K} \subset \mathbb{R}^d$ is a convex set of diameter $D$ and let $\{f(\cdot, x)\}_{x \in \Xi}$ be a family of convex functions over $\mathcal{K}$, which are $G$-Lipschitz and $\mu$-strongly convex. Given any sample set $S$ consists of $N$ samples from $\Xi$ and other necessary inputs, suppose we have a $(\varepsilon, \delta)$-DP algorithm $\mathcal{A}$ which can output a solution $\omega_T$ such that $\mathbb{E}[\widehat{F}(\omega_T) - \widehat{F}^*] = O \left( \frac{G^2 d \log(1/\delta)}{\mu \varepsilon^2 N^2} \right)$, where $\widehat{F}^* = \min_{\omega \in \mathcal{K}} \widehat{F}(\omega)$. Then when $\{h(\cdot, x)\}_{x \in \Xi}$ is only $G$-Lipschitz and convex with necessary inputs, for any sample set $S$ of size $N$, we also have a $(\varepsilon, \delta)$-DP algorithm $\mathcal{A}'$ which can get a solution $\omega_T$ such that $\mathbb{E}[\widehat{H}(\omega_T) - \widehat{H}^*] = O \left( \frac{GD \sqrt{d \log(1/\delta)}}{\varepsilon N} \right)$. where $\widehat{H}(\omega) = \frac{1}{N} \sum_{x_i \in S} h(\omega, x_i), \widehat{H}^* = \min_{\omega \in \mathcal{K}} H(\omega)$. The gradient complexity and privacy guarantee of $\mathcal{A}$ and $\mathcal{A}'$ are the same. Moreover, the reduction also holds for SCO.*

*Proof.* We only consider this lemma in the context of ERM, as we can use the nearly the same argument for SCO. Without loss of generality, we assume the intial point $\omega_0 = 0$. The proof of this reduction is rather simple: After getting $\{h(\cdot, x_i)\}_{x_i \in S}$, we only need to consider $h_u(\omega, x) = h(\omega, x) + u\|\omega\|^2$. Then $h_u(\cdot, x)$ is $u$-strongly convex and $O(G + uD)$-Lipschitz for any $x$ with $\|\omega\|_2 \leq D$ and $\omega \in \mathcal{K}$.

For the case $uD \leq G$, we run $\mathcal{A}$ on $\{h_u(\cdot, x_i)\}_{x_i \in S}$ to get a solution $\omega_T$ with loss $\mathbb{E}[H_u(\omega_T) - H_u^*] = O \left( \frac{G^2 d \log(1/\delta)}{u \varepsilon^2 N^2} \right)$, where $H_u(\omega) = \frac{1}{N} \sum_{x_i \in S} h(\omega, x_i) + u\|\omega\|^2$ and $H_u^* = \min_{\omega \in \mathcal{K}} H_u(\omega)$. Now by setting $u = \Theta \left( \frac{G \sqrt{d \log(1/\delta)}}{D \varepsilon N} \right)$, one has $\mathbb{E}[\widehat{H}(\omega_T) - \widehat{H}^*] = O \left( \frac{G^2 d \log(1/\delta)}{u \varepsilon^2 N^2} + u D^2 \right) = O \left( \frac{GD \sqrt{d \log(1/\delta)}}{\varepsilon N} \right)$. For the case $uD \geq G$, we have $\frac{GD \sqrt{d \log(1/\delta)}}{\varepsilon N} \geq GD$ and hence we can simply output the initial point $\omega_0$ as the solution with a loss no more than $GD$. □

The above reduction completes the main result of Theorem 1.1 for the general non-smooth case.

## 4 Differentially Private SCO

As mentioned before, for DP-SCO, we get the desired gradient complexity via an application of the iterative localization approach of Feldman et al [FKT20]. We prove the reduction theorem stated below, the proof of which is in the full version available as supplementary material.

**Theorem 4.1.** *Suppose we have an algorithm $\mathcal{A}$ which can solve ERM under strongly convex case and gets a solution with excess empirical loss $O(\frac{G^2}{\mu N})$ by using $g(N)$ many gradients, then we have*

an algorithm $\mathcal{A}'$ which can solve SCO under general case and gets a solution with excess population loss $O(\frac{GD}{\sqrt{N}})$ by using $\sum_{i=1}^{\lceil \log N \rceil} g(N/2^i)$ many gradients.

Moreover, for $\varepsilon, \delta \le 1/2$, if $\mathcal{A}_{\varepsilon,\delta}$ is $(\varepsilon, \delta)$-differentially private with excess empirical loss $O(\frac{G^2}{\mu}(\frac{1}{N} + \frac{d \log(1/\delta)}{\varepsilon^2 N^2}))$ under the strongly convex case by using $g(N, \varepsilon, \delta)$ many gradients, then we can get $(\varepsilon, \delta)$-differentially private $\mathcal{A}'$ with excess population loss $O(GD(\frac{1}{\sqrt{N}} + \frac{\sqrt{d \log(1/\delta)}}{\varepsilon N}))$ by querying gradients at most $\sum_{i=1}^{\lceil \log N \rceil} g(N/2^i, \varepsilon/2^i, \delta/2^i)$ times.

To obtain the optimal bounds for SCO, we first notice that we can afford to obtain sub-optimal bounds for ERM ($O(\frac{G^2}{\mu}(\frac{1}{N} + \frac{d \log(1/\delta)}{\varepsilon^2 N^2}))$), and hence fewer number of gradient queries. This observation combined with a careful choice parameters for ERM and the above theorem, helps us prove the main result in Theorem 1.2.

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
