}))$ while the lower bound of empirical risk is $\Omega(\frac{GD\sqrt{d}}{N\varepsilon})$, we do not know how to reduce from ERM to SCO.

We deviate from the above approaches for designing private algorithms for non-smooth functions. First notice that the gradient complexity $O(\frac{\varepsilon N^{\frac{3}{2}}}{d^{1/8} \log^{1/4}(1/\delta)} + \frac{\varepsilon^2 N^2}{d \log(1/\delta)})$ in Theorem 1.1 is the same for both strongly convex and general non-smooth functions; same holds for DP-SCO. This is not a coincidence; We prove that if we can achieve optimal empirical risk (population loss) for one case, then we can achieve optimal empirical risk (population loss) for another with the same privacy guarantee and gradient complexity. The Figure 1 illustrates the relationship among these different problems.

Our result for the general convex non-smooth case is obtained by providing a reduction to the strongly convex non-smooth case. Thus, our task becomes designing better algorithms for the strongly convex non-smooth functions. Rather than using SGD, we let the objective function take convolution with a sphere kernel to make it smooth. We then use the accelerated stochastic approximation algorithm in

[GL12] for solving strongly convex stochastic optimization problems. However, this is not enough, as the required noise that needs to be added to the gradients to make the algorithm private is too large to get subquadratic gradient complexity, even if we use the tighter Moments Accountant technique [ACG$^+$16]. We overcome this by increasing the batch size to an appropriate value. Combining these ideas together, we show that the amount of noise we add can be reduced to achieve the optimal excess empirical loss, and we get the gradient complexity of $O(\max\{N^{3/2}/d^{1/8}, N^2/d\})$.

For SCO, we get the gradient complexity of $O(\min\{N^{5/4}d^{1/8}, N^{3/2}/d^{1/8}\})$ via application of the iterative localization approach of Feldman et al [FKT20]. The intuition behind iterative localization is using private ERM to solve regularized objective functions which have low sensitivity, iteration by iteration. Each iteration reduces the distance to an approximate minimizer by a multiplicative factor, so after logarithmic number of phases we are done.

## 1.3 Concurrent and Independent Work

In an independent and concurrent work, [AFKT21] give a new analysis of private regularized mirror descent to do the private ERM. Then they combine the iterative localization approach to achieve the optimal excess population loss for SCO. Their result also achieves subquaratic gradient complexity.

More formally, they get $O(\log N \cdot \min(N^{3/2}\sqrt{\log d}, N^2/\sqrt{d}))$ for SCO in query complexity. We compare their gradient complexity with ours in the figure to right. As we see, our result is better in the important regime $d \leq N^{1+1/3}$. Same holds true for ERM. Finally, we remark that the main motivation of [AFKT21] was to study SCO problem in more general $\ell_p$ norms as much of the literature has focused on the $\ell_2$-norm. They also give new results in $\ell_p$-bounded domain together with another concurrent work [BGN21].

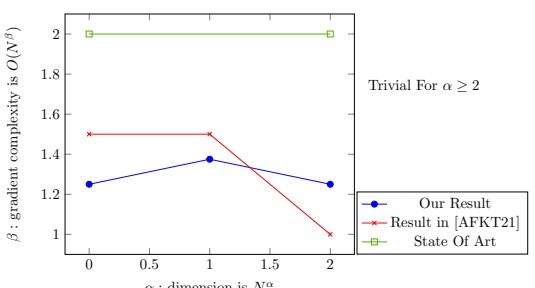

Figure 2: Comparison among our results, the recent result in [AFKT21] and the previous best one for the non-trivial regime ($d \leq N^2$). Suppose $\varepsilon, \delta$ are small constants. Our result is faster for the important case $d \leq N^{1+1/3}$.

## 2 Preliminaries

In this section, we briefly recall some of the main definitions we use from the convex optimization theory and differential privacy. We refer the readers to excellent books [Nes05, DR14] for more details on these topics.

### 2.1 Convex Optimization

**Definition 2.1** (Empirical risk minimization, Stochastic Convex Optimization). Let $\mathcal{K} \subset \mathbb{R}^d$ be a closed convex set of diameter $D$. Given a family of convex loss functions $\{f(\omega, x)\}_{x \in \Xi}$ of $\omega$ over $\mathcal{K}$ and a set of samples $S = \{x_1, \cdots, x_n\}$ over the universe $\Xi$, the objective of Empirical Risk Minimization (ERM) is to minimize

$$\widehat{F}(\omega) = \frac{1}{N} \sum_{x_i \in S} f(\omega, x_i).$$

The excess empirical loss with respect to a solution $\omega$ is defined by $\widehat{F}(\omega) - \widehat{F}^*$, where $\widehat{F}^* = \min_{\omega \in \mathcal{K}} \widehat{F}(\omega)$.

Stochastic Convex Optimization (SCO) wants to output a solution $\omega$ to minimize the expected loss (also referred to *population loss*) $F(\omega) - F^*$ where $F(\omega) = \mathbb{E}[x \sim \mathcal{P}]f(\omega, x)$ and $F^* = \min_{\omega \in \mathcal{K}} F(\omega)$.

**Definition 2.2** (L-Lipschitz Continuity). A function $f : \mathcal{K} \to \mathbb{R}$ is $L$-Lipschitz continuous over the domain $\mathcal{K} \subset \mathbb{R}^d$ if the following holds for all $\omega, \omega' \in \mathcal{K} : |f(\omega) - f(\omega')| \leq L\|\omega - \omega'\|_2$.

**Definition 2.3** ($\beta$-Smoothness). A function $f : \mathcal{K} \to \mathbb{R}$ is $\beta$-smooth over the domain $\mathcal{K} \subset \mathbb{R}^d$ if for all $\omega, \omega' \in \mathcal{K}$, $\|\nabla f(\omega) - \nabla f(\omega')\|_2 \leq \beta \|\omega - \omega'\|_2$.

**Definition 2.4** ($\mu$-Strongly convex). A differentiable function $f : \mathcal{K} \to \mathbb{R}$ is called strongly convex with parameter $\mu > 0$ if the following inequality holds for all points $\omega, \omega' \in \mathcal{K}$,

$$\langle \nabla f(\omega) - \nabla f(\omega'), \omega - \omega' \rangle \geq \mu \|\omega - \omega'\|_2^2.$$

Equivalently,

$$f(\omega') \geq f(\omega) + \nabla f(\omega)^\top (\omega' - \omega) + \frac{\mu}{2} \|\omega' - \omega\|_2^2.$$

### 2.2 Differential Privacy

**Definition 2.5** (Differential privacy). A randomized mechanism $\mathcal{M}$ is $(\varepsilon, \delta)$-differentially private if for any event $\mathcal{O} \in \mathrm{Range}(\mathcal{M})$ and for any neighboring databases that differ in a single data element, one has

$$\Pr[\mathcal{M}(S) \in \mathcal{O}] \leq \exp(\varepsilon) \Pr[\mathcal{M}(S') \in \mathcal{O}] + \delta.$$

**Lemma 2.6** (Proposition 2.1 in [DR14]). *(Post-Processing) Let $\mathcal{M} : \mathbb{N}^{|\Xi|} \to R$ be a randomized algorithm that is $(\varepsilon, \delta)$-differentially private. Let $f : R \to R'$ be an arbitrary randomized mapping. Then $f \circ \mathcal{M} : \mathbb{N}^{|\Xi|} \to R'$ is $(\varepsilon, \delta)$-differentially private.*

**Theorem 2.7** (Basic Composition). *Let $\mathcal{M}_i : \mathbb{N}^{|\Xi|} \to R_i$ be $(\varepsilon_i, \delta_i)$-differentially private. Then if mechanism $\mathcal{M}_{[k]} : \mathbb{N}^{|\Xi|} \to \prod_{i=1}^k \mathcal{R}_i$ is defined to be $\mathcal{M}_{[k]}(x) = (\mathcal{M}_1(x), \ldots, \mathcal{M}_k(x))$, then $\mathcal{M}_{[k]}$ is $(\sum_{i=1}^k \varepsilon_i, \sum_{i=1}^k \delta_i)$-differentially private.*

## 3 A Meta Algorithm for DP Convex Optimization

Many convex optimization algorithms with noisy first-order information have the following simple format.

---

**Algorithm 1:** Meta Algorithm META

---

1 **Input:** The objective convex function $F(\omega)$ we want to minimize, an initial point $\omega_0$.
2 **Process: for** *phases* $t = 1, \cdots,$ **do**
3 $\quad$ Get the noisy gradient $G_t \approx \nabla F(\omega_{t-1})$;
4 $\quad$ Update the result by some sub-procedure: $\omega_t \leftarrow$ Sub-procedure$(\omega_{t-1}, G_t)$;
5 **end**
6 **Output:** Some function of $\{\omega_i\}_{i \geq 1}$.

---

We can use the above algorithmic framework to solve ERM privately. Specifically, we make two simple modifications to make it private. First, we compute gradients over a uniform sample of some size $B$. Next, we add a carefully calibrated Gaussian noise to these gradients and take average, before updating our results. This gives us a meta differentially private algorithm for convex optimization problems, and is described in Algorithm 2. The DP analysis then follows from a careful accounting of the privacy budget lost in each iteration, and the bound on excess empirical risk comes from the property of the optimization algorithm.

---

**Algorithm 2:** Private Meta Algorithm META$_{\text{DP}}$

---

1 **Input:** Sample set $S = \{x_1, \cdots, x_N\}$, the objective convex function $F(\omega)$ we want to minimize, the initial point $\omega_0$, and privacy parameter $\varepsilon, \delta$;
2 **Process: for** *phases* $t = 1, \cdots, T$ **do**
3 $\quad$ Select a random sample set $S_t$ from the uniform distribution over all subsets of $S$ of size $B$;
4 $\quad$ Let $G_t = (\sum_{x_i \in S_t} \nabla f(\omega_{t-1}, x_i) + v)/B$, where $v \sim \mathcal{N}(0, \sigma^2 I_{d \times d})$;
5 $\quad$ Update the result by some sub-procedure $\omega_t \leftarrow$ Sub-procedure$(\omega_{t-1}, G_t)$;
6 **end**
7 **Output:** Some function of $\{\omega_i\}_{i \geq 1}$.

---

The above framework is a sub-sampled Gaussian mechanism, for which we can use tCDP proposed in [BDRS18] to analyze its privacy guarantee.

**Theorem 3.1.** *Suppose* $\{f(\cdot, x)\}_{x \in \Xi}$ *is a family of G-Lipschitz and convex functions over* $\mathcal{K}$*, for* $\varepsilon < c_1 B^2 T / N^2$*,* $B \le N/10$ *and* $1/2 \ge \delta > 0$*, by setting* $\sigma = \frac{c_2 G B \sqrt{T \log(1/\delta)}}{\varepsilon N}$ *for some constant* $c_1$ *and* $c_2$*,* META$_{\mathsf{DP}}$ *is* $(\varepsilon, \delta)$*-differential private.*

As mentioned before, we can use the result in [BDRS18] to give a formal proof of our result. Before we start, let us define something necessary.

**Definition 3.2** (Truncated CDP)**.** Let $\rho > 0$ and $\omega > 1$. A randomized algorithm $\mathcal{M} : \mathbb{N}^{|\Xi|} \to R$ satisfies $\omega$-truncated $\rho$-concentrated differential privacy (or $(\rho, \omega)$-tCDP) if for all neighboring $S, S'$ that differ in a single entry,

$$\forall \alpha \in (1, \omega), \mathrm{D}_\alpha \left( \mathcal{M}(S) \| \mathcal{M}(S') \right) \le \rho\alpha,$$

where $D_\alpha(\cdot \| \cdot)$ denotes the Rényi divergence [Rén61] of order $\alpha$ (in nats, rather than bits).

Similar to classic differential privacy, tCDP also enjoys a property of composition:

**Lemma 3.3** (Composition of tCDP)**.** *Let* $\mathcal{M}_1 : \mathbb{N}^{|\Xi|} \to R_1$ *satisfy* $(\rho, \omega)$*-tCDP and let* $\mathcal{M}_2 : \mathbb{N}^{|\Xi|} \times R_1 \to R_2$ *satisfy* $(\rho', \omega')$*-tCDP for all* $y \in R_1$*. Difine* $\mathcal{M} : \mathbb{N}^{|\Xi|} \to R_3$ *by* $\mathcal{M}(S) = \mathcal{M}_2(S, \mathcal{M}_1(S))$*. Then* $\mathcal{M}$ *satisfies* $(\rho + \rho', \min\{\omega, \omega'\})$*-tCDP.*

Now we state the main result of [BDRS18]:

**Theorem 3.4** (Privacy Amplification By Subsampling)**.** *Let* $\rho, s \in (0, 0.1]$ *and* $B, N \in \mathbb{N}$ *with* $q = B/N$ *and* $\log(1/q) \ge 3\rho(2 + \log_2(1/\rho))$*. Let* $\mathcal{M} : \mathbb{N}^{|\Xi|} \to R$ *satisfy* $(\rho, \omega')$*-tCDP for* $\omega' \ge \frac{\log(1/q)}{2\rho} \ge 3$*. Define the mechanism* $\mathcal{M}_q : \mathbb{N}^{|\Xi|} \to R$ *by* $\mathcal{M}_q(S) = \mathcal{M}(S_q)$ *where* $S_q \in \mathbb{N}^{|\Xi|}$ *is the restriction of* $S$ *to the entries specified by a uniformly ransom subset of size* $B$*.*

*The algorithm* $\mathcal{M}_q$ *satisfies* $(13q^2\rho, \omega)$*-tCDP for*

$$\omega = \frac{\log(1/q)}{4\rho}.$$

This theorem can apply to our algorithm META$_{\mathsf{DP}}$ directly, as we are using subsampling without replacement. More specifically, we are using subsampling Gaussian Mechanism, and for Gaussian Mechanism we have the following fact:

**Fact 3.5.** *Let* $P = \mathcal{N}(1, 1/2\rho)$ *and* $Q = \mathcal{N}(0, 1/2\rho)$*. Then* $D_\alpha(P \mid Q) = \rho\alpha$ *for all* $\alpha \in (1, \infty)$*. In other word, the Gaussian Mechanism with sensitive 1 satisfies* $(\rho, \infty)$*-tCDP.*

Now we are ready for our proof.

*Proof of Theorem 3.1.* For the $t$-th phase of META$_{\mathsf{DP}}$, let $\mathcal{M}(S) = \sum_{x \in S} \nabla f(\omega_{t-1}, x) + v$ where $v \sim \mathcal{N}(0, \sigma^2 I_{d \times d})$. As we are considering $G$-Lipschitz function $f$, then we know that $\|\nabla(f)\|_2 \le G$, which means that $\mathcal{M}$ is $(\rho, \infty)$-tCDP where $\rho = G^2/(2\sigma^2)$.

Assume our parameters satisfy the precondition of Theorem 3.4 first, then we know that the $t$-th phase of META$_{\mathsf{DP}}$ is $(13q^2\rho, 1/\rho)$-tCDP. By the composition property (Lemma 3.3), we know that META$_{\mathsf{DP}}$ is $(13Tq^2\rho, 1/\rho)$-tCDP.

When $Tq^2\rho \cdot \frac{\log(1/\delta)}{\varepsilon} \le O(\varepsilon)$ and $\frac{\log(1/\delta)}{\varepsilon} \le O(\frac{1}{\rho})$, we know that META$_{\mathsf{DP}}$ is $(\varepsilon, \delta)$-differentially private [BDRS18].

By setting $\sigma = \frac{c_2 G B \sqrt{T \log(1/\delta)}}{\varepsilon N}$, we have that $\rho = \frac{\varepsilon^2 N^2}{2c_2^2 B^2 T \log(1/\delta)}$. Together with the assumption $\varepsilon \le c_1 B^2 T / N^2$, we have both $Tq^2\rho \cdot \frac{\log(1/\delta)}{\varepsilon} \le O(\varepsilon)$ and $\frac{\log(1/\delta)}{\varepsilon} \le O(\frac{1}{\rho})$ as claimed. This completes the proof.

$\square$

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

Now we state the our modifications to make $\mathsf{AC-SA}$ private and prove its properties. Recall that $y \sim n_r$ is a $d$-dimension vector drawn from the uniform density on the $\ell_2$ ball of radius $r$. We start with the description of our algorithm.

---

**Algorithm 4:** Private $\mathsf{AC-SA}$

---

1 **Input:** A convex set $\mathcal{K}$ with diameter $D$, a family $\{f(\cdot, x_i)\}_{i \in [N]}$ of $G$-Lipschitz and $\mu$-strongly convex functions over $\mathcal{K}$, an initial point $\omega_0 \in \mathcal{K}$, privacy parameters $\varepsilon, \delta$, the batch size $B$, and the number of steps $T$.

2 Set $r \leftarrow \frac{D}{Td^{1/4}}$ and $\sigma \leftarrow \Theta(\frac{GB\sqrt{T \log(1/\delta)}}{\varepsilon N})$;

3 Run the $\mathsf{AC-SA}$ with the Oracle $\mathcal{G}$ defined below;

4 **Return:** The output of $\mathsf{AC-SA}$

5 **Oracle $\mathcal{G}(\omega)$:**

6 Select a random sample set $S_t$ from the uniform distribution over all subsets of $S$ of size $B$.

7 **Return:** $\left(\sum_{x_i \in S_t} \partial f(\omega + y_i, x_i) + v\right)/B$, where $y_i \sim n_r$ for each $i \in [B]$ and $v \sim \mathcal{N}(0, \sigma^2 \mathbf{I}_{d \times d})$.

---

### 4.1.3 Utility and Privacy

It is not hard to show that Private $\mathsf{AC-SA}$ (Algorithm 4) is an instance of $\mathsf{META_{DP}}$ (see Section 3), so we have the following guarantee directly by Theorem 3.1.

**Lemma 4.4.** *For $\varepsilon \leq c_1 B^2 T/N^2, \delta \leq 1/2, B \leq N/10$ and $\sigma = \frac{c_2 GB\sqrt{T \log(1/\delta)}}{\varepsilon N}$ where $c_1 \leq 1, c_2 \geq 1$ are constants, Private $\mathsf{AC-SA}$ is $(\varepsilon, \delta)$-DP.*

Now we consider the accuracy of Private $\mathsf{AC-SA}$.

**Lemma 4.5.** *Under the assumptions defined in Algorithm Private $\mathsf{AC-SA}$, after $T$ iterations, it outputs $\omega_T$ such that*

$$\mathbb{E}[\widehat{F}(\omega_T) - \widehat{F}^*] = O\left(\frac{G^2/B + \sigma^2 d/B^2}{\mu T} + \frac{GDd^{1/4}}{T}\right),$$

*where $\omega^* = \arg\min_{\omega \in \mathcal{K}} \widehat{F}(\omega)$, and $\widehat{F}^* = \min_\omega \widehat{F}(\omega)$.*

*Proof.* By Claim 4.2, we know that $\widehat{F}_{n_r}$ is $G$-Lipschitz and $\frac{G\sqrt{d}}{r}$-smooth. Furthermore, by Fact 4.3, we know that $\widehat{F}_{n_r}$ is $\mu$-strongly convex. For any $t$th iteration, one has that $\mathbb{E}[\mathcal{G}_t] = \nabla \widehat{F}_{n_r}(\omega_t^{md})$ and $\mathbb{E}[\|\mathcal{G}_t - \nabla \widehat{F}_{n_r}(\omega_t^{md})\|_*^2] \leq G^2/B + \sigma^2 d/B^2$. Then by Theorem 4.1 with $M = 0, L = \frac{G\sqrt{d}}{r}, V = G^2/B + \sigma^2 d/B^2$, we get

$$\mathbb{E}[\widehat{F}_{n_r}(\omega_T) - \min_\omega \widehat{F}_{n_r}(\omega)] = O\left(\frac{G^2/B + \sigma^2 d/B^2}{\mu T} + \frac{GD^2\sqrt{d}}{T^2 r}\right).$$

Next, by the first bullet of Claim 4.2, we know that $\widehat{F}(\omega) \leq \widehat{F}_{n_r}(\omega) \leq \widehat{F}(\omega) + Gr$ for any $\omega$. Combining these together, we get

$$\begin{aligned}
&\mathbb{E}[\widehat{F}(\omega_T) - \widehat{F}(\omega^*)] \\
&= \mathbb{E}[\widehat{F}(\omega_T) - \widehat{F}_{n_r}(\omega_T)] + \mathbb{E}[\widehat{F}_{n_r}(\omega_T) - \min_\omega \widehat{F}_{n_r}(\omega)] + \min_\omega \widehat{F}_{n_r}(\omega) - \widehat{F}(\omega^*) \\
&\leq 2Gr + O(\frac{G^2/B + \sigma^2 d/B^2}{\mu T} + \frac{GD^2\sqrt{d}}{T^2 r}).
\end{aligned}$$

By setting $r = \frac{Dd^{1/4}}{T}$, we completes the proof. $\qquad\square$

Before stating the main result of this section, we prove two technical lemmas that can remove the dependence on the diameter term. Lemma 4.6 below is used to prove Lemma 4.7.

**Lemma 4.6.** *Consider a sequence $x_1, x_2, \cdots$. Suppose $0 \leq x_1 \leq n$ and $0 \leq x_{i+1} \leq \sqrt{x_i} + 1$, then for $k \geq \lceil \log\log n \rceil$, one has that $x_k \leq 16$.*

*Proof.* Without loss of generality, let $x_{i+1} = \sqrt{x_i} + 1$.

We construct another sequence $y_1, \cdots, y_k$ such that $y_1 = x_1$ and $y_{i+1} = 2\sqrt{y_i}$. Then by induction, it is easy to prove that for each $i \in [k]$, $y_i \geq x_i$. So we only need to prove that $y_k \leq 16$.

Let $z_i = \log_2 y_i$, then one has $z_{i+1} = z_i/2 + 1$. Obviously, we know that $z_i = 2^{-i+1}(z_1 - 2) + 2$ and $z_k \leq 4$, which means that $x_k \leq y_k \leq 16$. $\qquad\square$

Recall that the lower bound of strongly convex case is $\Omega(\min\{\frac{G^2}{\mu}, \frac{G^2 d\log(1/\delta)}{\mu\varepsilon^2 N^2}\})$ while for the general case is $\Omega(\min\{GD, \frac{GD\sqrt{d\log(1/\delta)}}{\varepsilon N}\})$. Therefore, we only need to think about the case when $\frac{d\log(1/\delta)}{\varepsilon^2 N^2} \leq 1$, or the bound will be trivial. The following lemma says if we can achieve sum of these two lower bounds for strongly-convex case, then we can achieve the optimal bound for the strongly-convex case, which implies we can reduce the Strongly-Convex Case to General Convex Case. The following lemma follows from the reduction in Section 5.1 in [FKT20], and we try to give a more formal statement for convenience in the future.

**Lemma 4.7** (Reduction to General Convex Case). *Given $\widehat{F}$ is $G$-Lipschitz and $\mu$-strongly convex. Suppose for any $\varepsilon, \delta < 1/2$, we have an $(\varepsilon, \delta)$-differentially private algorithm $\mathcal{A}$ which takes $\omega_0$ as the initial start point and outputs a solution $\omega_T$ such that*

$$\mathbb{E}[\widehat{F}(\omega_T) - \widehat{F}^*] = O\left(\frac{G^2 d\log(1/\delta)}{\mu\varepsilon^2 N^2} + \frac{GD\sqrt{d\log(1/\delta)}}{\varepsilon N}\right),$$

*where $\omega^* = \arg\min_{\omega\in\mathcal{K}} \widehat{F}(\omega)$ and $D = \|\omega_0 - \omega^*\|_2$. Then by taking $\mathcal{A}$ as sub-procedure with some modifications on parameters, we can get an $(\varepsilon, \delta)$-differentially private solution with excess empirical loss at most*

$$\mathbb{E}[\widehat{F}(\omega_T) - \widehat{F}^*] = O\left(\frac{G^2 d\log(1/\delta)}{\mu\varepsilon^2 N^2}\right).$$

*Furthermore, if $\mathcal{A}$ uses $g(N, \varepsilon, \delta)$ many gradients, the new algorithm uses $\sum_{i\geq 1} g(N, \varepsilon/2^i, \delta/2^i)$ many gradients.*

*Remark* 4.8. All algorithms in this paper uses less gradients if $\varepsilon$ and $\delta$ are smaller. So, the new algorithm uses essentially as much as the given algorithm.

*Proof.* Repeat the private algorithm $\mathcal{A}$ for $k = \lceil \log \log N^3 \rceil$ times. For the $i$th repetition, we start from the output of the last repetition and use $\mathcal{A}$ as a sub-procedure with privacy parameter $\varepsilon_i = \varepsilon/2^{k+1-i}$ and $\delta_i = \delta/2^{k+1-i}$. (Note that the noise is decreasing so that the last step gives the best solution). We show that the last output has excess empirical risk at most $O\left(\frac{G^2 d \log(1/\delta)}{\mu \varepsilon^2 N^2}\right)$.

More specifically, let $\omega_i$ be the output of the $i$th repetition, $\Delta_i = \mathbb{E}[\widehat{F}(\omega_i) - \widehat{F}^*]$ and $D_i^2 = \mathbb{E}[\|\omega_i - \omega^*\|^2]$. As the objective function $\widehat{F}$ is $\mu$-strongly convex, we know that $\frac{1}{2}\mu D_i^2 \le \Delta_i$ for all $i \ge 0$.

By the guarantee of the algorithm, there exists some constant $c \ge 1$ such that

$$
\begin{aligned}
\Delta_{i+1} &= \mathbb{E}[\widehat{F}(\omega_{i+1}) - \widehat{F}^*] \\
&\le c\frac{GD_i\sqrt{d\log(1/\delta_i)}}{\varepsilon_i N} + c\frac{G^2 d \log(1/\delta_i)}{\mu \varepsilon_i^2 N^2} \\
&\le c\frac{G\sqrt{d\log(1/\delta_i)}}{\varepsilon_i N}\sqrt{\frac{2\Delta_i}{\mu}} + \frac{E_i}{c},
\end{aligned}
$$

where we define $E_i = 2c^2 \frac{G^2 d \log(1/\delta_i)}{\mu \varepsilon_i^2 N^2}$.

As $E_i/E_{i+1} = \frac{\varepsilon_{i+1}^2 \log(1/\delta_i)}{\varepsilon_i^2 \log(1/\delta_{i+1})} \le 8$, we can rearrange the above function and get

$$
\begin{aligned}
\frac{\Delta_{i+1}}{64E_{i+1}} &\le \frac{\sqrt{\Delta_i E_i} + \frac{E_i}{c}}{64E_{i+1}} \\
&\le \frac{E_i}{64E_{i+1}}\left(\sqrt{\frac{\Delta_i}{E_i}} + \frac{4}{c}\right) \\
&\le \sqrt{\frac{\Delta_i}{64E_i}} + 1.
\end{aligned}
$$

By strong convexity one has that $\Delta_1 \le G^2/\mu$, and $E_1 = \Omega(G^2 \log^3 N/(\mu N^2)) = \Omega(G^2/(\mu N^3))$ by the definition, so $\Delta_1/E_1 \le N^3$. Then by Lemma 4.6, after $k = \lceil \log \log N^3 \rceil$ repetitions, we get $\frac{\Delta_k}{64E_k} \le 16$. This further implies that there is a solution with expected error

$$
\mathbb{E}[\widehat{F}(\omega_k) - \widehat{F}^*] = O(\frac{G^2 d \log(1/\delta)}{\mu \varepsilon^2 N^2}).
$$

The privacy guarantee comes directly from the basic composition theorem (See Theorem 2.7). $\quad\square$

We did not optimize constants in the calculations above. Now we are ready to state the main result for the strongly-convex case.

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

## 5 Differentially Private SCO

In this section we study SCO. We can use the iterative localization technique in [FKT20] to reduce the SCO problem to an ERM problem. More specifically, if we can solve private ERM and get a (nearly) optimal empirical loss, then we can solve private SCO with (nearly) optimal excess population loss with the following algorithm framework (Algorithm 5). See Theorem 5.1 for the corresponding formal statement.

---

**Algorithm 5:** Iterative Localized Algorithm Framework $\mathcal{A}'$

---

1 **Input:** A family of $G$-Lipschitz and $\mu$-strongly convex function $f : \mathcal{K} \times \Xi \to \mathbb{R}$, initial point $\omega_0 \in \mathcal{K}$ and privacy parameter $\varepsilon, \delta$.
2 **Process:** Set $k = \lceil \log N \rceil$;
3 **for** $i = 1, \cdots, k$ **do**
4      Set $\varepsilon_i = \varepsilon/2^i, N_i = N/2^i, \eta_i = \eta/2^{5i}$;
5      Apply $(\varepsilon_i, \delta_i)$-DP ERM algorithm $\mathcal{A}_{\varepsilon_i, \delta_i}$ over $\mathcal{K}_i = \{\omega \in \mathcal{K} : \|\omega - \omega_{i-1}\|_2 \le 2G\eta_i N_i\}$
     with the function $\widehat{F}_i(\omega) = \frac{1}{N_i} \sum_{j \in S_i} f(\omega, x_j) + \frac{1}{\eta_i N_i} \|\omega - \omega_{i-1}\|^2$ where $S_i$ consists of
     $N_i$ samples with replacement from $\mathcal{P}$;
6      Let $\omega_i$ be the output of the ERM algorithm;
7 **end**
8 **Return:** The final iterate $\omega_k$;

---

**Theorem 5.1.** *Suppose we have an algorithm $\mathcal{A}$ which can solve ERM under strongly convex case and gets a solution with excess empirical loss $O(\frac{G^2}{\mu N})$ by using $g(N)$ many gradients, then we have an algorithm $\mathcal{A}'$ which can solve SCO under general case and gets a solution with excess population loss $O(\frac{GD}{\sqrt{N}})$ by using $\sum_{i=1}^{\lceil \log N \rceil} g(N/2^i)$ many gradients.*

*Moreover, for $\varepsilon, \delta \le 1/2$, if $\mathcal{A}_{\varepsilon, \delta}$ is $(\varepsilon, \delta)$-differentially private with excess empirical loss $O(\frac{G^2}{\mu}(\frac{1}{N} + \frac{d \log(1/\delta)}{\varepsilon^2 N^2}))$ under the strongly convex case by using $g(N, \varepsilon, \delta)$ many gradients, then we can get $(\varepsilon, \delta)$-differentially private $\mathcal{A}'$ with excess population loss $O(GD(\frac{1}{\sqrt{N}} + \frac{\sqrt{d \log(1/\delta)}}{\varepsilon N}))$ by querying gradients at most $\sum_{i=1}^{\lceil \log N \rceil} g(N/2^i, \varepsilon/2^i, \delta/2^i)$ times.*

We only prove the bound with privacy guarantee, as the (non-private) bound can be proved with similar argument. Two technical lemmas will be proved at first, after which we will complete the proof.

**Lemma 5.2.** *Let $\widehat{\omega}_i = \arg \min_{\omega \in \mathcal{K}} \widehat{F}_i(\omega)$, then*

$$\mathbb{E}[\|\omega_i - \widehat{\omega}_i\|_2^2] \le O(\frac{G^2 \eta_i^2 d \log(1/\delta_i)}{\varepsilon_i^2} + G^2 \eta_i^2 N_i).$$

*Proof.* At first, we prove that $\widehat{\omega}_i \in \mathcal{K}_i$. The definition of $\widehat{\omega}_i$ implies that

$$\frac{1}{N_i} \sum_{j=1}^{N_i} f(\widehat{\omega}_i, x_j) + \frac{1}{\eta_i N_i} \|\widehat{\omega}_i - \omega_{i-1}\|_2^2 \le \frac{1}{N_i} \sum_{j=1}^{N_i} f(\omega_{i-1}, x_j).$$

Then we know that

$$\frac{1}{\eta_i N_i} \|\widehat{\omega}_i - \omega_{i-1}\|_2^2 \le G\|\widehat{\omega}_i - \omega_{i-1}\|_2,$$

which implies $\widehat{\omega}_i \in \mathcal{K}_i$.

Next, note that $\widehat{F}_i$ is $\lambda_i = \frac{1}{\eta_i N_i}$-strongly convex, by the guarantee of our ERM algorithm, we know that

$$\begin{aligned}
\frac{\lambda_i}{2} \mathbb{E}[\|\widehat{\omega}_i - \omega_i\|_2^2] &\le \mathbb{E}[\widehat{F}_i(\widehat{\omega}_i) - \widehat{F}_i(\omega_i)] \\
&\le O(\frac{G^2 d \log(1/\delta_i)}{\lambda_i \varepsilon_i^2 N_i^2} + \frac{G^2}{\lambda_i N_i}) \\
&= O(\frac{G^2 \eta_i d \log(1/\delta_i)}{\varepsilon_i^2 N_i} + G^2 \eta_i),
\end{aligned}$$

which implies

$$\mathbb{E}[\|\widehat{\omega}_i - \omega_i\|_2^2] \leq O(\frac{G^2\eta_i^2 d\log(1/\delta_i)}{\varepsilon_i^2} + G^2\eta_i^2 N_i).$$

$\square$

**Lemma 5.3.** *For any $y \in \mathcal{K}$, we know that*

$$\mathbb{E}[F(\widehat{\omega}_i) - F(y)] \leq \frac{\mathbb{E}[\|\omega_{i-1} - y\|_2^2]}{\eta_i N_i} + O(G^2\eta_i).$$

*Proof.* Let $r(\omega, x) = f(\omega, x) + \frac{1}{\eta_i N_i}\|\omega - \omega_{i-1}\|_2^2$, $R(\omega) = \mathbb{E}_{x\sim\mathcal{P}} r(\omega, x)$ and $y^* = \arg\min_{\omega\in\mathcal{K}} R(\omega)$. By Theorem 6 in [SSSSS09], one has that

$$\begin{aligned}
\mathbb{E}[R(\widehat{\omega}_i) - R(y)] &= \mathbb{E}[F(\widehat{\omega}_i) + \frac{1}{\eta_i N_i}\|\widehat{\omega}_i - \omega_{i-1}\|_2^2 - F(y) - \frac{1}{\eta_i N_i}\|y - \omega_{i-1}\|_2^2] \\
&\leq \mathbb{E}[R(\widehat{\omega}_i) - R(y^*)] \\
&\leq O(G^2\eta_i),
\end{aligned}$$

which implies that

$$\begin{aligned}
\mathbb{E}[F(\widehat{\omega}_i) - F(y)] &\leq O(G^2\eta_i) - \frac{1}{\eta_i N_i}\mathbb{E}[\|\widehat{\omega}_i - \omega_{i-1}\|_2^2] + \frac{1}{\eta_i N_i}\mathbb{E}[\|y - \omega_{i-1}\|_2^2] \\
&\leq O(G^2\eta_i) + \frac{1}{\eta_i N_i}\mathbb{E}[\|y - \omega_{i-1}\|_2^2].
\end{aligned}$$

$\square$

Having these two lemmas, we can begin the proof.

*Proof of Theorem 5.1.* The privacy guarantee comes directly from the basic composition theorem (See Theorem 2.7).

Let $S_i = \{x_j\}_{N-N/2^{i-1}\leq j\leq N-N/2^i}$. Let $N_i = N/2^i, \varepsilon_i = \varepsilon/2^i, \delta_i = \delta/2^i$ and $\eta_i = \eta/2^{5i}$ where $\eta$ will be defined soon. For $i \in [k]$, let $\widehat{F}_i(\omega) = \sum_{x_j\in S_i} f(\omega, x_j) + \frac{1}{\eta_i N_i}\|\omega - \omega_{i-1}\|_2^2$.

Let $\widehat{\omega}_0 = \omega^*$, we have

$$\mathbb{E}[F(\omega_k)] - F(\omega^*) = \sum_{i=1}^k \mathbb{E}[F(\widehat{\omega}_i) - F(\widehat{\omega}_{i-1})] + \mathbb{E}[F(\omega_k) - F(\widehat{\omega}_k)].$$

First, Lemma 5.2 implies that

$$\begin{aligned}
\mathbb{E}[F(\omega_k) - F(\widehat{\omega}_k)] &\leq O(G\sqrt{\mathbb{E}[\|\omega_k - \widehat{\omega}_k\|_2^2]}) \\
&\leq O(\frac{G^2\eta_k\sqrt{d\log(1/\delta_k)}}{\varepsilon_k} + G^2\eta_k\sqrt{N_k}) \\
&= O(\frac{G^2\eta\sqrt{d\log(N/\delta)}}{\varepsilon N^3} + \frac{G^2\eta}{N^4}),
\end{aligned}$$

which is negligible.

Then one has

$$\begin{aligned}
\sum_{i=1}^k \mathbb{E}[F(\widehat{\omega}_i) - F(\widehat{\omega}_{i-1})] &\leq \sum_{i=1}^k \frac{\mathbb{E}[\|\widehat{\omega}_{i-1} - \omega_{i-1}\|_2^2]}{\eta_i N_i} + O(G^2\eta_i) \\
&\leq O(\frac{D^2}{\eta N} + \eta G^2 + \sum_{i=2}^k (\frac{G^2\eta_i d\log(1/\delta_i)}{\varepsilon_i^2 N_i} + G^2\eta_i))
\end{aligned}$$

$$\leq O\left(\frac{D^2}{\eta N} + \eta G^2 + \frac{G^2 \eta d \log(1/\delta)}{\varepsilon^2 N}\right).$$

By setting $\eta = \frac{D}{G} \cdot \min\{\frac{1}{\sqrt{N}}, \frac{\varepsilon}{\sqrt{d\log(1/\delta)}}\}$, we get the excess population loss:

$$\mathbb{E}[F(\widehat{\omega}_k) - F(\omega^*)] = O\left(GD\left(\frac{1}{\sqrt{N}} + \frac{\sqrt{d\log(1/\delta)}}{N\varepsilon}\right)\right).$$

As for the gradient complexity, as we use $g(N_i, \varepsilon_i, \delta_i)$ queries of gradients in $i$-th iteration, the total gradient complexity is $\sum_{i=1}^{k} g(N_i, \varepsilon_i, \delta_i)$ as claimed. $\qquad\square$

Note that Theorem 5.1 allows the ERM algorithm has an extra $G^2/(\mu N)$ loss. This allows us to design a faster ERM algorithm compared Theorem 4.9 by choosing a different set of parameters.

**Lemma 5.4.** *Under the assumption defined in Algorithm Private* AC$-$SA*, with*

$$O\left(N + \min\{\sqrt{\varepsilon}N^{5/4}d^{1/8}, \frac{\varepsilon N^{\frac{3}{2}}}{d^{1/8}\log^{1/4}(1/\delta)}\}\right)$$

*gradient complexity, one can get a solution $\omega_T$ such that*

$$\mathbb{E}[\widehat{F}(\omega_T) - \widehat{F}^*] = O\left(\frac{G^2}{\mu}\left(\frac{1}{N} + \frac{d\log(1/\delta)}{\varepsilon^2 N^2}\right)\right).$$

*Proof.* By Lemma 4.5, one has

$$\mathbb{E}[\widehat{F}(\omega_T) - \widehat{F}^*] = O\left(\frac{G^2/B + \sigma^2 d/B^2}{\mu T} + \frac{GDd^{1/4}}{T}\right).$$

Again, setting $\sigma = \frac{c_2 GB\sqrt{T\log(1/\delta)}}{\varepsilon N}$ one has

$$\mathbb{E}[\widehat{F}(\omega_T) - \widehat{F}^*] = O\left(\frac{G^2}{\mu BT} + \frac{G^2 d\log(1/\delta)}{\mu\varepsilon^2 N^2} + \frac{GDd^{1/4}}{T}\right).$$

Taking $T = 400\lceil\min\{N^{1/2}d^{1/4}, \frac{N\varepsilon}{d^{1/4}\sqrt{\log(1/\delta)}}\}\rceil$ and using $BT \geq N$ (which we will ensure), we have

$$\mathbb{E}[\widehat{F}(\omega_T) - \widehat{F}^*] = O\left(\frac{G^2}{\mu N} + \frac{G^2 d\log(1/\delta)}{\mu\varepsilon^2 N^2} + \frac{GD}{\sqrt{N}} + \frac{GD\sqrt{d\log(1/\delta)}}{\mu\varepsilon N}\right)$$

$$= O\left(\frac{G^2}{\mu}\zeta + GD\sqrt{\zeta}\right).$$

where $\zeta = \frac{1}{N} + \frac{d\log(1/\delta)}{\varepsilon^2 N^2}$.

To ensure that Private AC$-$SA is $(\varepsilon, \delta)$-DP, we set $B = \lceil N/T + N\sqrt{\varepsilon/T}\rceil = \Theta(N/T + N\sqrt{\varepsilon/T})$. By our choice of $T$, we have $B \leq N/10$ and $\varepsilon \leq c_1 B^2 T/N^2$. Hence, we can apply Lemma 4.4 to conclude $(\varepsilon, \delta)$-DP.

Hence, we have a $(\varepsilon, \delta)$-DP for ERM with loss $O\left(\frac{G^2}{\mu}\zeta + GD\sqrt{\zeta}\right)$ with $\zeta = \frac{1}{N} + \frac{d\log(1/\delta)}{\varepsilon^2 N^2}$. Note however that Theorem 5.1 requires us to have a DP-ERM algorithm with loss $O(\frac{G^2}{\mu}\zeta)$, namely, we have the extra term $O(GD\sqrt{\zeta})$. To remove this term, we follow the reduction in Lemma 4.7.

We note that the exact same proof as Lemma 4.7 shows that for any $\zeta > 0$, if we can solve strongly ERM with loss $O(G^2\zeta^2/\mu + GD\zeta)$, then we can solve strongly ERM with loss $O(G^2\zeta^2/\mu)$ by using the same number of gradient. This completes the proof.

$\qquad\square$

Before stating our result on SCO, we need the following variant of Lemma 4.7. The proof is essentially the same, we state it for future reference.

**Lemma 5.5** (Reduction to General Convex Case). *Given $F$ is $G$-Lipschitz and $\mu$-strongly convex. Suppose for any $\varepsilon, \delta < 1/2$, we have an $(\varepsilon, \delta)$-differentially private algorithm $\mathcal{A}$ which takes $\omega_0$ as the initial start point and $N$ samples i.i.d drawn from some distribution $\mathcal{P}$, and outputs a solution $\omega_T$ such that*

$$\mathbb{E}[F(\omega_T) - F^*] = O\left(\frac{G^2}{\mu}(\frac{1}{N} + \frac{d\log(1/\delta)}{N^2\varepsilon^2}) + GD(\frac{1}{\sqrt{N}} + \frac{\sqrt{d\log(1/\delta)}}{N\varepsilon})\right),$$

*where $\omega^* = \arg\min_{\omega\in\mathcal{K}} F(\omega)$ and $D = \|\omega_0 - \omega^*\|$. Then by taking $\mathcal{A}$ as sub-procedure with some modifications on parameters, we can get an $(\varepsilon, \delta)$-differentially private solution with excess population loss at most*

$$\mathbb{E}[F(\omega_T) - F^*] = O\left(\frac{G^2}{\mu}(\frac{1}{N} + \frac{d\log(1/\delta)}{N^2\varepsilon^2})\right).$$

*Furthermore, if $\mathcal{A}$ uses $g(N, \varepsilon, \delta)$ many gradients, the new algorithm uses $\sum_{i\geq 1} g(N/2^i, \varepsilon/2^i, \delta/2^i)$ many gradients.*

*Proof.* The only difference to Lemma 4.7 is that this algorithm takes $N/2^{k+1-i}$ samples instead of $N$ samples in the $i$-th step for $k = \lceil \log\log N^3 \rceil$, so it may have less gradient complexity. The rest of the proof is identical. $\square$

Now, we can get the result for general convex case by Theorem 5.1 and Lemma 5.4, then extend it to strongly convex case by Lemma 5.5.

**Theorem 5.6** (DP-SCO, Theorem 1.2 restated). *Suppose $\varepsilon, \delta \leq 1/2$. Let $\{f(\cdot, x)\}_{x\in\Xi}$ is convex and $G$-Lipschitz with respect to $\ell_2$ norm and convex over $\mathcal{K}_r$, where $r = \frac{D\sqrt{d\log(1/\delta)}}{\varepsilon N}$, there is an $(\varepsilon, \delta)$-differentially private algorithm which takes*

$$O(N + \min\{\sqrt{\varepsilon}N^{5/4}d^{1/8}, \frac{\varepsilon N^{3/2}}{d^{1/8}\log^{1/4}(1/\delta)}\})$$

*gradient queries to get a solution $\omega_T$*

$$\mathbb{E}[F(\omega_T) - F(\omega^*)] = O(GD(\frac{1}{\sqrt{N}} + \frac{\sqrt{d\log(1/\delta)}}{N\varepsilon}).$$

*Moreover, if $\{f(\cdot, x)\}_{x\in\Xi}$ is also $\mu$-strongly convex over $\mathcal{K}_r$, we can use the same gradient complexity and get a solution $\omega$ such that:*

$$\mathbb{E}[F(\omega_T) - F(\omega^*)] = O\left(\frac{G^2}{\mu}(\frac{d\log(1/\delta)}{\varepsilon^2 N^2} + \frac{1}{N})\right).$$