# OpenReview forum: "Private Non-smooth ERM and SCO in Subquadratic Steps"
_NeurIPS.cc/2021/Conference — NeurIPS 2021 Spotlight_

### Official Review · Reviewer_qrbc · 2021-07-10

**Rating:** 7
**Confidence:** 4

**Summary:**

This paper addresses the question of the gradient complexity of differentially private (DP) convex empirical risk minimization (ERM) and stochastic convex optimization (SCO). It provides substantial improvements compared to most work in the area, and it provides the best running times on some ranges of parameters.

The main results in this work are:

1. An Accelerated Stochastic Gradient method with randomized smoothing, and therefore applicable to nonsmooth losses (Lemmata 3.4 and 3.5)
2. A two-way reduction between convex and strongly convex ERM (Lemmata 3.6 and 3.7)
3. A two-way reduction between convex and strongly convex SCO (Lemmata 3.7 and 5.5)

These results are used in combination to show that both DP-ERM and DP-SCO can be solved to optimal rates with subquadratic number of gradient evaluations (with respect to sample size). This resolves an open question from a recent work of Bassily et al. (BFGT20). On the other hand, a concurrent work by Asi et al. (AFKT21) provides subquadratic algorithms in the same setting, but the current submission provides faster rates when $d\leq N^{4/3}$ (this range is interesting, as when $d$ gets closer to $N^2$ the rates for DP-ERM and DP-SCO become vacuous.


**Main Review:**

The results of this paper are definitely interesting for the NeurIPS community. Arguably, there is not huge novelty in the methods used, but the idea of accelerated randomized smoothing seems novel in the context of differential privacy (this idea has been used in non private SCO and online learning).

I now elaborate further on my comments above:

- Regarding contributions 2 and 3 stated in my previous point, they seem to follow quite directly from the cited work FKT20. This leads to my comment on please being careful on citing the preceding work: It seems to me that Lemma 4.7 in the supplementary follows quite closely Thm 5.1 in FKT20 (the arXiv version).

- Unfortunately, the concurrent work AFKT21 is already capable of obtaining major improvements in complexity. However, the current submission points out that that is not the end of the story, and currently the complexity of DP-ERM and DP-SCO looks quite complex.

Finally, I would like to make some minor comments and questions:

- Does AC-SA provide optimal rates for smooth DP-ERM? If so, what is its complexity? Even if this problem has already been solved (with linear complexity) I thought it could be interesting to consider methods based on acceleration.

- Have the authors tried a higher sampling rate for the randomized smoothing? Right now, a single stochastic evaluation is used, so I was wondering if decreasing the variance of this randomization by using more random directions could improve the complexity.

- In the introduction the term 'universal convergence' is used. I believe 'uniform convergence' is a much more standard term, so I would recommend adopting it. In a related note, the work Fel16 proves lower bounds for uniform convergence in stochastic convex optimization, whereas the paper says 'not necessarily convex.' That should be corrected.

**Time Spent Reviewing:**

3

---

> ### Author Response · Authors · 2021-08-10
> **Reply to Reviewer 3**
>
> Thanks for your careful review and suggestion. We appreciate your positive feedback. Now we address your specific comments.
>
> Regarding contributions 2 and 3.  Thanks for your suggestions regarding citing results in FKT20.
> We think that the reductions in 2 and 3 are somewhat standard by now. We included formal proof of them only for completeness and convenience.  In the next version, we will be more careful and point out our new ideas and what already follows from FKT20.
>
> Regarding AFKT21 and the complexity of DP-ERM and SCO. We agree with you that AFKT21 is also capable of beating quadratic gradient complexity. However, both our work and AFKT21 are not the final answers to this problem, and there may be DP-ERM and SCO algorithms with linear gradient complexity. We believe that this is an exciting and important open problem.
>
> Does AC-SA provide optimal rates for smooth DP-ERM? This is a great question. If we assume the smoothness of the objective function, then we can skip the randomized smoothing. But we think that this alone cannot get the linear gradient complexity. We will address this question in the next version of the paper.
>
> Have the authors tried a higher sampling rate for the randomized smoothing?  This is a great suggestion, and we have not tried this. But our intuition says that it cannot work out directly. Intuitively, more evaluations can decrease the variance, but larger noise is required to keep it private. We need to calculate this tradeoff more carefully to see if your idea can lead to better bounds.
>
> Universal convergence Vs Uniform convergence: Thanks so much for pointing this out. We agree with you, and we will fix it in the next version.

---

> > ### Comment · Reviewer_qrbc · 2021-08-13
> > **answer to authors response**
> >
> > Thank you for the responses

---

### Official Review · Reviewer_fSmY · 2021-07-18

**Rating:** 7
**Confidence:** 4

**Summary:**

The paper studies the empirical risk minimization and stochastic convex optimization problems under differential privacy. Previous works give private algorithms for ERM that achieve nearly-optimal utility in the convex and Lipschitz as well as the strongly convex settings but have gradient complexity that is at least quadratic in the number of samples. Thus a natural open question that has been raised in prior works is whether one can obtain an algorithm with a sub-quadratic gradient complexity. The current paper answers this question affirmatively in the regime where the dimension is a super-constant function of the number of samples. Using techniques developed in prior work, the paper extends the ERM algorithm to the more general SCO problem.


**Limitations And Societal Impact:**

Yes

**Main Review:**

Overview of the approach/techniques: The main result for ERM is obtained by carefully putting together techniques from convex optimization as well as differential privacy. The algorithm follows the general template of DP algorithms, and the privacy analysis follows readily from prior works. The improvement in the running time comes from utilizing faster convex optimization algorithms as well as a careful setting of the parameters such as batch size to control the noise added to ensure privacy. Specifically, the algorithm uses the standard reduction from the general convex to the strongly convex setting where one adds a strongly convex quadratic term to the objective. In order to leverage acceleration, the algorithm uses a technique from prior work that smoothens the objective via a convolution with a sphere kernel. The resulting objective is now smooth and strongly convex (with some appropriate smoothness and strong convexity), and one can employ an accelerated gradient descent algorithm to optimize it. The algorithm uses the ACSA algorithm and its analysis as a black box for this purpose. After putting together these pieces, the main task is to carefully analyze the utility/convergence tradeoff and find suitable settings for the parameters such as the batch size.

Significance: The paper addresses fundamental problems in optimization with privacy. The paper shows for the first time that sub-quadratic running time is possible and paves the way towards further improvements. On the negative side, the approach seems to be primarily a proof of concept that faster running time is possible by piecing together existing techniques, rather than an approach that could eventually lead to the ideal nearly-linear running time.

Novelty/originality: Although there is some novelty to the overall approach, the main results are obtained via a careful combination of existing techniques. The paper does not introduce any substantial new techniques.

Quality: Overall this is a solid result and a valuable contribution to differentially private optimization.

Clarity: The clarity could be improved by fixing typos. The paper could be made more readable by providing signposting in the technical sections and some further guidance and intuition for the proofs.


**Time Spent Reviewing:**

I did not track the hours

---

> ### Author Response · Authors · 2021-08-10
> **Reply to Reviewer 1**
>
> Thank you for the careful review and suggestions. We appreciate your positive feedback. As suggested, we will provide more clear signposting and add intuitions behind our proofs in the next version.

---

### Official Review · Reviewer_45fX · 2021-08-27

**Rating:** 8
**Confidence:** 4

**Summary:**

This paper studies the problem of differentially private ERM and SCO for non-smooth function, aiming mainly to improve the gradient query complexity of previous algorithms. The authors develop algorithms for DP-SCO and ERM that achieve the optimal rates with a subquadratic qradient query complexity which improves the complexity of prior work. The algorithms are based on smoothening the function via randomized smoothing, then applying a noisy accelerated gradient algorithms for smooth functions.

**Limitations And Societal Impact:**

Yes

**Main Review:**

The paper provides nice progress on the problems of DP-ERM and DP-SCO where the authors obtain improved gradient query complexity compared to prior work. Prior work has mainly obtained complexity $O(n^2)$ and this papers achieves $O(n^{1+3/8})$ when $d=n$ (which is currently the best known complexity for this setting). Moreover, the algorithmic techniques are interesting therefore I recommend to accept the paper.

Some of the techniques in the paper has appeared in prior work and this should be mentioned. For example, the combination of smoothening and accelerated algorithms has appeared in [DBW12] in the (non-private) optimization literature. The algorithms in this paper can therefore be seen as a noisy variant of the algorithm in [DBW12] (similarly to noise SGD). Moreover, some of the reductions in the paper has also appeared in prior and concurrent work. The reduction from strongly convex DP-SCO to convex DP-SCO appeared in [FKT20] and the reduction from DP-SCO to DP-ERM appeared in [AFKT21].



Other comments:

1. The paper is in general clearly-written though some theorem statements could be further improved. For example, the first part of theorem 4.1 can be removed as it is not a crucial part of the theorem. Other theorem statements can be shortened too.

2. In algorithm 3, the randomized smoothing noise (y in line 7) should not be the same for different indices $i$. This is important to enable the variance calculations in the proof (line 303 in appendix).

3. Typo in line 23 in the definition of $F(w)$

4. Maybe use $n$ instead of $N$ for sample size as common in DP optimization literature.

5. line 54: It was actually [FKT20] that first gave $O(n^2)$ gradient complexity for non-smooth functions (not [BFGT20]).

6. Theorem 3.1: the theorem statement says "there is an algorithm that has" but I guess this should be "algorithm 2 has".

7. Typo in lower bounds in lines 249 and 250.

8. Reduction to convex case (lemma 3.6): doesn't this result follow directly from the reductions of [FKT20] (section 5.1)?





**Time Spent Reviewing:**

1.5

---

### Decision · Program_Chairs · 2021-09-27

**Decision:**

Accept (Spotlight)

**Comment:**

This paper proposes a new algorithm for differentially private stochastic convex optimization and ERM with non-smooth convex losses. Previous algorithms for the problem required a quadratic number of gradient computations. This work uses a combination of a new idea and existing techniques in optimization to reduce the number of steps to essentially $N^{11/8}$. This is a significant progress on a fairly basic question in DP optimization. Therefore I recommend acceptance.